# Mechanically activated *snai1b* coordinates the initiation of myocardial delamination for trabeculation

Jing Wang [1], Aaron L. Brown [2], Seul-Ki Park[3,4], Charlie Z. Zheng [5], Adam Langenbacher[5], Enbo Zhu [3,4], Ryan O'Donnell[3], Peng Zhao [3,4], Jeffrey J. Hsu[3], Tomohiro Yokota[3,4], Jiandong Liu [6], Jau-Nian Chen [5], Alison L. Marsden[7] & Tzung K. Hsiai [1,3,4] ✉

During development, myocardial contractile force and intracardiac hemodynamic shear stress coordinate the initiation of trabeculation. While Snail family genes are well-recognized transcription factors of epithelial-to-mesenchymal transition, *snai1b*-positive cardiomyocytes are sparsely distributed in the ventricle of zebrafish at 4 days post-fertilization. Isoproterenol treatment significantly increases the number of *snai1b*-positive cardiomyocytes, of which 80% are Notch-negative. CRISPR-activation of *snai1b* leads to 51.6% cardiomyocytes forming trabeculae, whereas CRISPR-repression reduces trabecular cardiomyocytes to 6.7% under isoproterenol. In addition, 36.7% of *snai1b*-repressed cardiomyocytes undergo apical delamination. 4-D strain analysis demonstrates that isoproterenol increases the myocardial strain along radial trabecular ridges in alignment with the *snai1b* expression and Notch-ErbB2-mediated trabeculation. Single-cell and spatial transcriptomics reveal that these *snai1b*-positive cardiomyocytes are devoid of some epithelial-to-mesenchymal transition-related phenotypes, such as Col1a2 production and induction by ErbB2 or TGF-β. Thus, we uncover *snai1b*-positive cardiomyocytes that are mechanically activated to initiate delamination for cardiac trabeculation.

During cardiac morphogenesis, mechanical cues coordinate the initiation of ventricular trabeculation and valve formation[1–6]. The myocardium differentiates into two layers: an outer compact zone and an inner trabeculated zone. The muscular ridges within the ventricles of the heart, known as trabeculae organize into a network of branching outgrowths from the myocardial wall, which provide blood perfusion before the coronary vasculature fully develops and are essential for myocardial contraction[7]. A reduction in trabeculation is observed in congenital heart defects, such as ventricular compact zone deficiencies, where hypertrabeculation (non-compaction) is associated with left ventricular non-compaction (LVNC)[8]. LVNC is considered to be the third most common cardiomyopathy after dilated and hypertrophic cardiomyopathy in the pediatric population[9]. Its prevalence was estimated from 4.5 to 26 per 10,000 adult patients referred for echocardiographic

[1]Department of Bioengineering, University of California, Los Angeles, Los Angeles, CA, USA. [2]Department of Mechanical Engineering, Stanford University, Stanford, CA, USA. [3]Division of Cardiology, Department of Medicine, School of Medicine, University of California, Los Angeles, Los Angeles, CA, USA. [4]Department of Medicine, Greater Los Angeles Veteran Affairs Healthcare System, Los Angeles, CA, USA. [5]Department of Molecular, Cell, and Developmental Biology, University of California, Los Angeles, Los Angeles, CA, USA. [6]Department of Pathology and Laboratory Medicine, McAllister Heart Institute, University of North Carolina at Chapel Hill, Chapel Hill, NC, USA. [7]Departments of Pediatrics and Bioengineering, Stanford University, Stanford, CA, USA. ✉e-mail: thsiai@mednet.ucla.edu

diagnosis[10]. However, the mechanisms of mechanotransduction underlying the initiation of trabecular organization remain elusive[8,11].

In zebrafish, trabeculation is initiated as a coordinated interplay between biochemical and mechanical signaling, where myocardial contractile force and hemodynamic shear coordinate the lateral activation vs. inhibition of Notch-Nrg1-ErbB2 signaling to regulate myocardial delamination and proliferation for trabeculation[2,4,12–15]. The arrest of cardiomyocyte contraction either by troponin T type 2a (*tnnt2a*) inhibition or in weak atrium[m58] (*wea*) mutants led to a reduction in ventricular or atrial contractility, respectively, and subsequent attenuation in hemodynamic shear stress-mediated Notch signaling to initiate trabeculation[15]. Crossbreeding of the *Tg(flk:mCherry)* line with the *Tg(TP1:EGFP)* Notch reporter line revealed shear stress-mediated endocardial Notch activation. Deletion of endocardium in the *cloche*[sk4] mutants significantly downregulated cardiac Notch signaling, leading to non-trabeculated ventricles[15]. These findings support the notion that hemodynamic shear stress activates endocardial Notch signaling-mediated trabeculation, essential for contractile function during cardiac development[2].

Trabecular organization or patterning further modulates ventricular remodeling and cardiac strain. Temporal variation in hemodynamic shear stress, namely, time-averaged wall shear stress (TAWSS) and oscillatory shear index (OSI), coordinates the organization of trabecular ridges and grooves[2,16,17]. In wild-type zebrafish, pulsatile shear stress activates endocardial Notch along the trabecular ridges at 3 days post-fertilization (dpf), whereas oscillatory shear stress reduces endocardial Notch activity in the trabecular grooves at 4 dpf. In silico simulation demonstrated that trabeculation promotes kinetic energy (KE) dissipation, whereas the non-trabeculated ventricle lacks KE dissipation, resulting in reduced myocardial contractile force and subsequent ventricular remodeling[2]. We also reported that changes in intracardiac hemodynamic force in the non-trabeculated ventricle were associated with increased ventricular volume and reduced axial strain[2]. In addition to providing oxygen diffusion, these findings suggest that trabeculation enhances both myocardial contraction and KE dissipation.

Myocardial contractile force is also essential to initiate valvular formation, while hemodynamic shear stress contributes to valve leaflet growth. Increased hemodynamic shear stress activates Notch1b-mediated endothelial-mesenchymal transition (EndoMT) to promote ventriculobulbar (VB, i.e., OFT) valve formation, whereas decreased contractility attenuates Notch1b-activated EndoMT[3]. Snail family genes, such as *Snai1* (*SNAI1* in humans), are well-recognized transcription factors for epithelial-to-mesenchymal transition (EMT) during valvulogenesis[18,19], tissue fibrosis[20], and regeneration[21]. Other mechanosensitive transcription factors, including Nfatc1, regulate the spatiotemporal expression of Snail genes within valvular endocardial cells to coordinate their differentiation and migration[22–25]. Nevertheless, whether Snail activation in the ventricle is implicated in trabecular organization is unknown.

In this context, we sought to demonstrate that the *snai1b*-positive (⁺) cardiomyocytes (CMs) are mechanically activated to initiate delamination for cardiac trabeculation. Increasing evidence suggests that one of the two zebrafish *Snai1* genes, *snai1b*, guides the migration of myocardial progenitors at the initial stage of cardiac development[26,27]. We revealed that the number of *snai1b*⁺ CMs is increased by 10-fold in response to elevated myocardial strain, where 80% of the *snai1b*⁺ CMs were Notch-negative and undergoing delamination for trabeculation. The distribution of *snai1b*⁺ CMs spatially aligned with the radial trabecular strain. Moreover, the myocardial *snai1b* activation is independent of endocardial shear stress and Notch-mediated ErbB2 signaling. Furthermore, CRISPR-activation or inhibition experiments confirmed that myocardial *snai1b* mediates the initiation of trabecular delamination. Single-cell and spatial transcriptomics revealed the absence of some canonical EMT pathways such as *col1a2* and TGF-β in these *snai1b*⁺ CMs. Thus, we uncover the mechano-

activated *snai1b*-positive but Notch-negative cardiomyocytes to coordinate the initiation of delamination for trabecular organization.

## Results

### *snai1b*-postive cardiomyocytes

From 56 h to 6 days post-fertilization (hpf), *snai1b* was sparsely expressed in the embryonic myocardium in the transgenic *Tg(snai1-b:EGFP; myl7:mCherry)* reporter line[28], but was prominent in the bulbus-ventricular (BV) annulus region (Fig. 1a, b, d). The *snai1b:EGFP* signal also colocalized with the CM nuclei in the *Tg(snai1b:EGFP; myl7:mCherry-zCdt1)* embryos[29] (Fig. 1c). At 4 dpf, *snai1b* mRNA (in situ hybridization) was detected in the epicardium and atrioventricular/outflow tract (AV/OFT) valve leaflets (Supplementary Fig. 1), as previously reported[23,30,31]. However, the colocalization of *snai1b* mRNA with the EGFP reporter signal was solely observed in CMs, where 93% of all EGFP⁺ CMs were mRNA⁺ (Supplementary Fig. 2). This could be due to the reporter construct not capturing epicardial or endocardial enhancers within its promoter sequence[28].

### Mechano-sensitive *snai1b*-positive cardiomyocytes

Isoproterenol (ISO, an β₁ and β₂ receptor agonist) was administered at 1 dpf to increase myocardial contraction and ventricular strain, as evidenced by both heart rate and ejection fraction from 48 to 96 hpf (Supplementary Fig. 5a). ISO treatment significantly increased the number of *snai1b*⁺ ventricular CMs during trabeculation (Fig. 1d, e and Supplementary Fig. 1). At 14 dpf, *snai1b*⁺ CMs remained sparsely distributed in the ventricle (Fig. 2a). Under ISO treatment, they developed into a continuous trabecular network (treatment to 11 dpf) (Fig. 2b).

At 4 dpf, we performed 4-D strain mapping on the in vivo images of *Tg(myl7:mCherry)* hearts acquired via spinning disk confocal microscopy. The mapping revealed an ISO-mediated increase in the average "area strain"[32] (stretching of the myocardium) (Fig. 2c, d) and spatial variations in the area strain over a cardiac cycle (Fig. 2e, see the ensuing Fig. 3). The increase is more accentuated across the endocardial surface than epicardial surface, suggesting that the trabecular layer exhibits greater ISO-induced changes in strain than the compact layer. Moreover, ISO-treatment did not change the number of compact and trabecular CMs, or the size of compact and trabecular cross-sectional area, indicating that the overall development of ventricle was unaffected (Supplementary Fig. 3). However, the size of ventricle increased (Supplementary Fig. 3b, d), and the thinning of trabecular, not compact, layer was observed, which suggests an increase of trabecular wall stress[33].

From 1 to 5 dpf, co-treating the ISO group with myosin inhibitors, para-amino-blebbistatin (pAB, 10 μM) or 2,3-butanedione 2-monoxime (BDM, 10 mM)[34,35], decreased the temporal and spatial variations in myocardial strain, and the frequency of observing *snai1b*⁺ CMs was reduced to 84.6% and 50%, respectively (Supplementary Fig. 4a, b). When ISO and pAB/BDM treatment were delayed to 3 dpf (72 hpf), the number of *snai1b*-EGFP⁺ CMs was significantly lower (Supplementary Fig. 4c). However, the number of *snai1b*-mRNA⁺ CMs was not reduced, except for the ISO-pAB group. Taken together, these findings suggest that ISO-mediated increase in ventricular strain is implicated in activating myocardial *snai1b*. Next, we aimed to elucidate the spatial variation in strain patterns and distribution of *snai1b*⁺ CMs in the trabecular network.

### Myocardial strain is in alignment with the radial trabecular ridges

At 4 dpf, ISO-induced myocardial strain aligned with the 3-D trabecular organization (Fig. 3a, b). The trabecular ridges formed radially from the AV canal region to the ventricular outer curvature, while some ridges developed circumferentially to connect to the radial ridges. ISO treatment increased the strain (shortening) along radial ridges during systole by 31.6% (*p* = 0.0013 vs. control, n = 6 ridges for each group), without

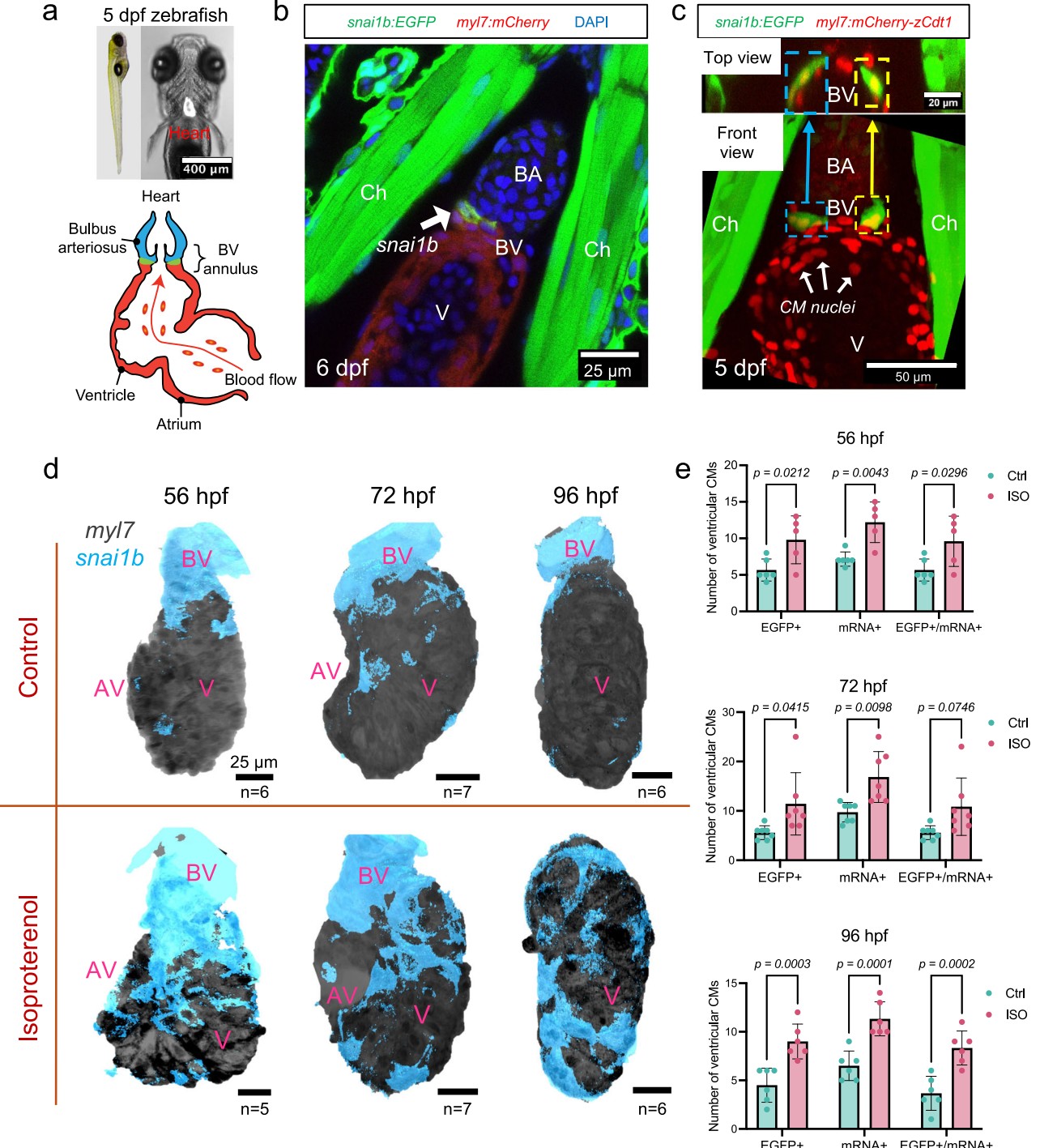

**Fig. 1 | Myocardial *snai1b* expression at the BV annulus and in the ventricle during trabeculation. a** A zebrafish embryo (left panel) at 5 dpf, and its heart is highlighted. Right panel illustrates the two-chambered cardiac anatomy. **b** At 6 dpf, confocal imaging of hearts of *Tg(snai1b:EGFP; myl7:mCherry)* larvae. The intensity of *snai1b* expression (green, arrow) was highest at the BV annulus (n = 5). **c** At 5 dpf, confocal imaging of hearts from *Tg(myl7:mCherry-zCdt1)* embryos (n = 7) demonstrates that *snai1b* fluorescence (colored squares) was colocalized to CM nuclei. Upper panel provides a view of the BV annulus from the BA. **d** During trabeculation (56–96 hpf), *snai1b:EGFP* signal is concentrated around the BV annulus. Isoproterenol (ISO) treatment at 1 dpf induced *snai1b* activation in the ventricular CMs. **e** Whole-mount in situ hybridization of *snai1b* mRNA in the *Tg(snai1b:EGFP; myl7:mCherry)* reporter line reveals a significant increase of both EGFP[+] and mRNA[+] CMs in ISO-treated hearts. (see Supplementary Figs. 1 and 2 for staining images). All values are displayed with mean and standard deviation (SD). *p*-value is displayed for each comparison. The number of hearts analyzed in each is displayed in (**d**). Ordinary two-way ANOVA followed by Šídák's multiple comparisons test on the means was applied to determine statistical significance. Source data are provided as a Source Data file. Anatomic labels: BA bulbus arteriosus, V ventricle, BV bulbus-ventricular annulus, Ch chest wall, AV atrioventricular canal.

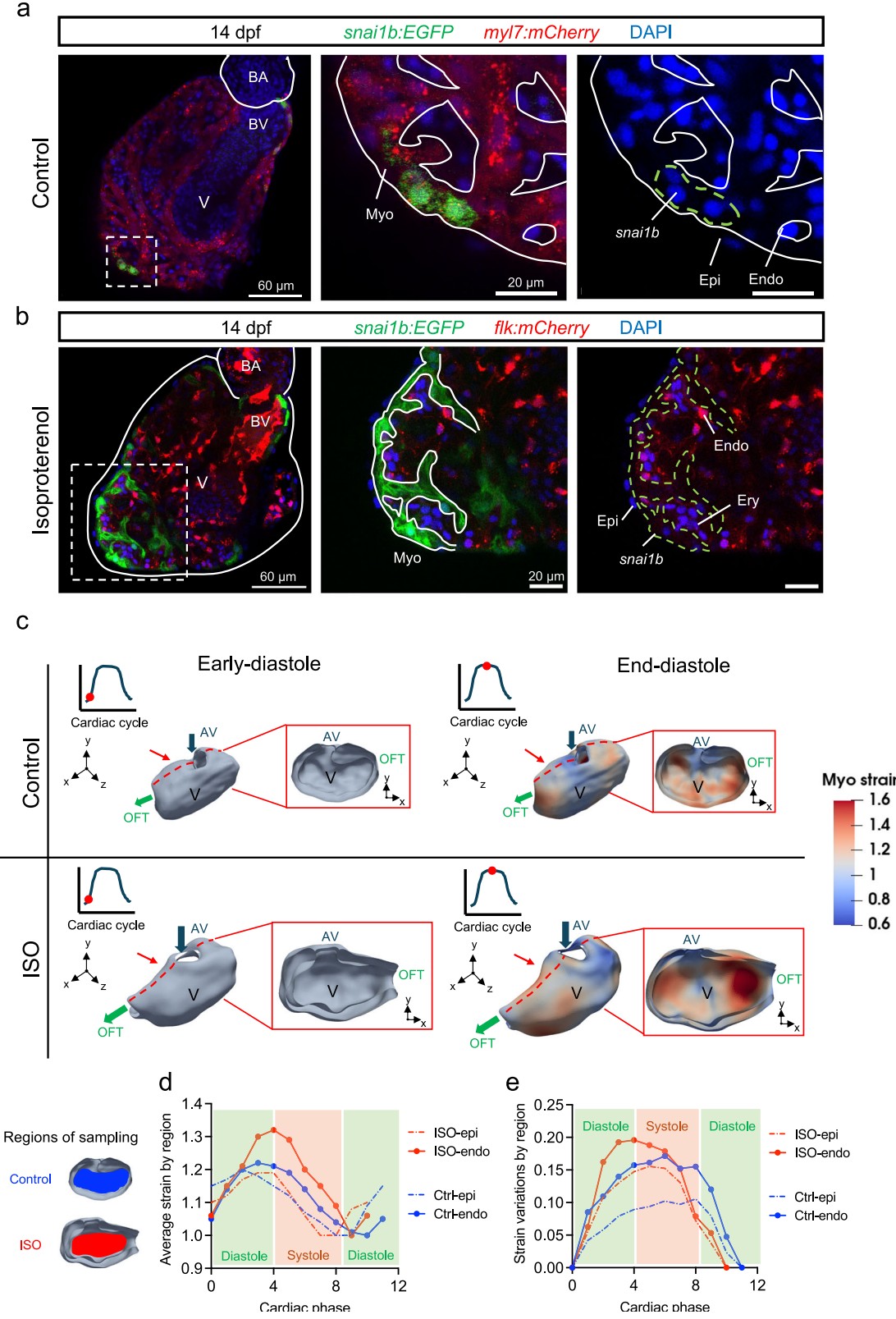

**a** 14 dpf    *snai1b:EGFP*    *myl7:mCherry*    DAPI

**b** 14 dpf    *snai1b:EGFP*    *flk:mCherry*    DAPI

**c** Early-diastole / End-diastole — Control / ISO; Myo strain scale 0.6–1.6

**d** Regions of sampling (Control / ISO); Average strain by region vs Cardiac phase

**e** Strain variations by region vs Cardiac phase

affecting the circumferential ridges (Fig. 3b). 2-D cross sections revealed that the radial trabecular ridges underwent transmural thickening (elongation) during systole, while the surrounding compact (cortical) layer underwent circumferential contraction (Fig. 3c). ISO treatment resulted in a 136% increase of transmural strain in the radial trabecular ridges ($p < 0.0001$ vs. control, n = 6 ridges for control, n = 7 for ISO), and the circumferential strain in the compact layer remained statistically unchanged (Fig. 3d). Further, we measured the changes in cross-

sectional area using a *Tg(myl7:mKate-CAAX)* membrane line as CMs contracted at both 56 and 96 hpf (Supplementary Fig. 5b–e). Our data showed that ISO increased the deformation of trabecular but not compact CMs at both time points. The change in the cross-sectional area can be an estimate of strain because CMs are considered incompressible with a conserved volume during contraction[36]. These results are consistent with a recent study demonstrating that trabecular CMs experience higher strain along its fiber direction than the compact CMs[37].

**Fig. 2 | ISO-mediated increase in ventricular contractility and strain activates myocardial *snai1b*. a, b** At 14 dpf, the expression of myocardial *snai1b* (dashed green outline) remained sparse in the ventricle, whereas Isoproterenol (ISO) treatment from 1 to 11 dpf revealed a persistent *snai1b* activation in the trabecular network. Anatomic labels: BA bulbus arteriosus, V ventricle, BV bulbus-ventricular annulus, Myo myocardium, Endo endocardium, Epi epicardium, Ery erythrocyte. **c** At 4 dpf, 4-D mapping of myocardial strain in a control and an ISO-treated heart. Two time points during diastole were displayed, and the ISO-treated heart experienced higher strain at the end-diastole time point than the control. Red dashed lines and squares mark the cross-section planes, and the red arrows indicate the viewing direction. Blue and green arrows indicate the direction of blood flow. Anatomic labels: V ventricle, AV atrioventricular canal, OFT outflow tract. **d** Average epicardial/endocardial ventricular strain within the sampling regions during one cardiac cycle. ISO treatment increased the myocardial strain across the endocardial surface during the end-diastolic and systolic phases. Source data are provided as a Source Data file. **e** Strain variations within the sampling regions, calculated as the standard deviation of strains at each time point (phase). ISO treatment induced greater myocardial strain variations compared with the control heart. Source data are provided as a Source Data file.

Thus, ISO treatment increased the strain aligning with radially organized trabecular ridges, giving rise to the spatial variation in strain in alignment with the distribution of *snai1b*+ CMs. A recent report indicates that local variation of cellular tension triggers myocardial delamination during trabeculation[4]. For this reason, we sought to elucidate whether ISO-activated *snai1b*+ CMs initiate delamination for trabecular organization.

### *snai1b*+/Notch− cardiomyocytes initiate delamination

During the initiation of trabeculation, lateral activation and inhibition were implicated in the trabecular organization into ridges and grooves. Hemodynamic shear stress activates endocardial Notch-ephrinB2-neuregulin (Nrg) signaling to the ErbB2 receptors in the delaminating CMs[15]. However, activation of Notch in the neighboring CMs inhibits the ErbB2 signaling to inhibit their delamination[12–14].

We performed in situ hybridization of *snai1b* mRNA in *Tg(TP1:EGFP)* embryos, a Notch reporter line[38]. In control hearts, trabecular CMs were devoid of Notch, with Notch-positive CMs adjacent to the trabeculae (Fig. 4a, d), whereas in ISO-treated hearts, *snai1b* activation occurred in Notch-negative CMs in both compact (cortical) and trabecular layers (Fig. 4b and Supplementary Fig. 6a, b). At 4 dpf, 75% of control (n = 6 embryos) and 83% of ISO-treated *snai1b*+ CMs (n = 7 embryos) were Notch-negative (Fig. 4f). A small percentage (3.4%) of *snai1b*+ CMs underwent apical delamination in ISO-treated hearts, also surrounded by Notch-positive CMs (Supplementary Fig. 6c). These results reveal that *snai1b*+/Notch− CMs are involved in initiating myocardial delamination.

A prior study has shown that mechanical signaling is epistatic to Nrg/ErbB2 signaling and drives myocardial delamination without ErbB2 activation[4]. We applied a selective ErbB2 inhibitor, PD168393 (PD), to ISO-treated *Tg(TP1:EGFP)* embryos starting at either 55 hpf or 72 hpf (hours post-fertilization). Though PD treatment impaired trabeculation, the total number of *snai1b*+ CMs per heart was unaffected (*p* > 0.05, n = 5 for 55 hpf group, n = 7 for 72 hpf group) (Fig. 4c, e and Supplementary Fig. 6d, e). The percentage of Notch-negative *snai1b*+ CMs remained statistically unchanged (87.7% for 55 hpf group, 83% for 72 hpf group), while the trabecular *snai1b*+ population was reduced from 50% to 37% and to 39%, respectively (Fig. 4f). Furthermore, PD treatment significantly reduced the total number of Notch+ CMs, while ISO alone did not (Supplementary Fig. 7). This result is consistent with a reported experiment where PD treatment downregulated several Snail genes, except for *snai1b*, in the heart at 4 dpf[14], and it further supports that *snai1b* is activated by myocardial contractile force.

### Activation of myocardial *snai1b* initiates delamination

To corroborate *snai1b*-mediated delamination, we utilized a CM-specific Tol2-CRISPR activation/interference (CRISPRa/i) system[39], whereby Tol2 transposon plasmids were injected in one-cell stage embryos to induce mosaic activation or repression of *snai1b* among CMs (Fig. 5a and Supplementary Fig. 8). We quantified the location of control (injected with scramble sgRNAs), *snai1b*-repressed, and *snai1b*-activated CMs with and without ISO treatment. At 4 dpf, following ISO treatment, *snai1b*-activation significantly increased the percentage of delaminated (Supplementary Fig. 9b) and trabecular CMs per heart (Fig. 5b, c). 51.6% of

total *snai1b*-activated CMs developed into trabeculae, compared to 25% of control and 6.7% of *snai1b*-repressed CMs with ISO (Fig. 5d). While the majority of control (58.3%) and *snai1b*-repressed (56.7%) CMs remained in the compact layer, 36.7% of *snai1b*-repressed CMs delaminated apically following ISO treatment. This phenotype is consistent with an earlier report that *snai1b* knockout caused apical extrusion of CMs at 50 hpf[27], suggesting the role of *snai1b* in the initiation of myocardial delamination. Without ISO, *snai1b*-activation led to a significant increase in trabecular CMs, while *snai1b*-repression did not affect the percentage of trabecular CMs (Fig. 5c and Supplementary Fig. 9a). This result further demonstrated that *snai1b* coordinates proper delamination for trabeculation under elevated cardiac strain.

### Enrichment analysis of mesenchymal genes in *snai1b*+ CMs

To unravel the genes associated with *snai1b*+ CMs, we analyzed the recently published single-cell RNA sequencing (scRNA-seq) dataset of zebrafish hearts at 5 dpf[40]. Consistent with our data based on the reporter line and in situ hybridization, *snai1b*+ CMs were rare among the sequenced cells (4 out of 366), and the *snai1b*+ population was primarily present in epicardial, valvular interstitial (VICs), and bulbus smooth muscle (SMCs) cells (Supplementary Fig. 10a–g). We isolated the CMs and the SMCs using their marker genes, *myl7* and *elnb*/*mylka*, to construct a new dataset (Fig. 6a). Interestingly, the new dataset self-reorganized into three clusters: a ventricular cluster with only CMs, a bulbus arteriosus cluster with mostly SMCs, and a BV annulus cluster with both CMs and SMCs (Fig. 6b). The *snai1b*+ CMs belonged to the BV annulus cluster, consistent with our in vivo finding (see Fig. 1b). Three of the four *snai1b*+ CMs also expressed epicardial or VIC markers (Supplementary Fig. 10h, i). These could be due to cross-contamination during sequencing or a transient cell identity during the embryonic stage[41].

Next, we performed Gene Ontology (GO) enrichment analysis to uncover the biologically correlated genes and pathways within BV annulus and ventricular CMs (Fig. 6c–f). BV annulus CMs had enriched mesenchymal genes not found in the ventricular CMs, such as *tgfbr2b*, *mdka*, and *igf2b*[42], and a number of extracellular matrix (ECM) genes were enriched in these BV annulus CMs, including collagens, *dcn*, and *lama5*, further suggesting an EMT-like state[43]. Unlike ventricular CMs, BV annulus CMs featured a host of downregulated genes involved in sarcomeric structure, calcium handling, and electron transport machinery, indicating a shift of phenotype in their actomyosin network (Supplementary Fig. 11).

### Canonical pathways in *snai1b*+ CMs vs. epicardial and valvular cells

As ECM remodeling is an essential aspect of EMT regulated by Snail genes[19,20,26], we performed in situ hybridization of *col1a2* (collagen 1a2) mRNA at 4 dpf (Fig. 7a–d). While *col1a2* was predicted to be the most upregulated ECM gene, we did not find *snai1b*+ CMs expressing *col1a2* in the control, ISO-treated, or CRISPRa-injected hearts. Nevertheless, colocalization between *col1a2* and *snai1b* mRNA could be clearly seen in epicardial cells and VICs.

We also tested the function of TGF-β signaling in regulating myocardial *snai1b*, which is a well-recognized activator of Snail genes

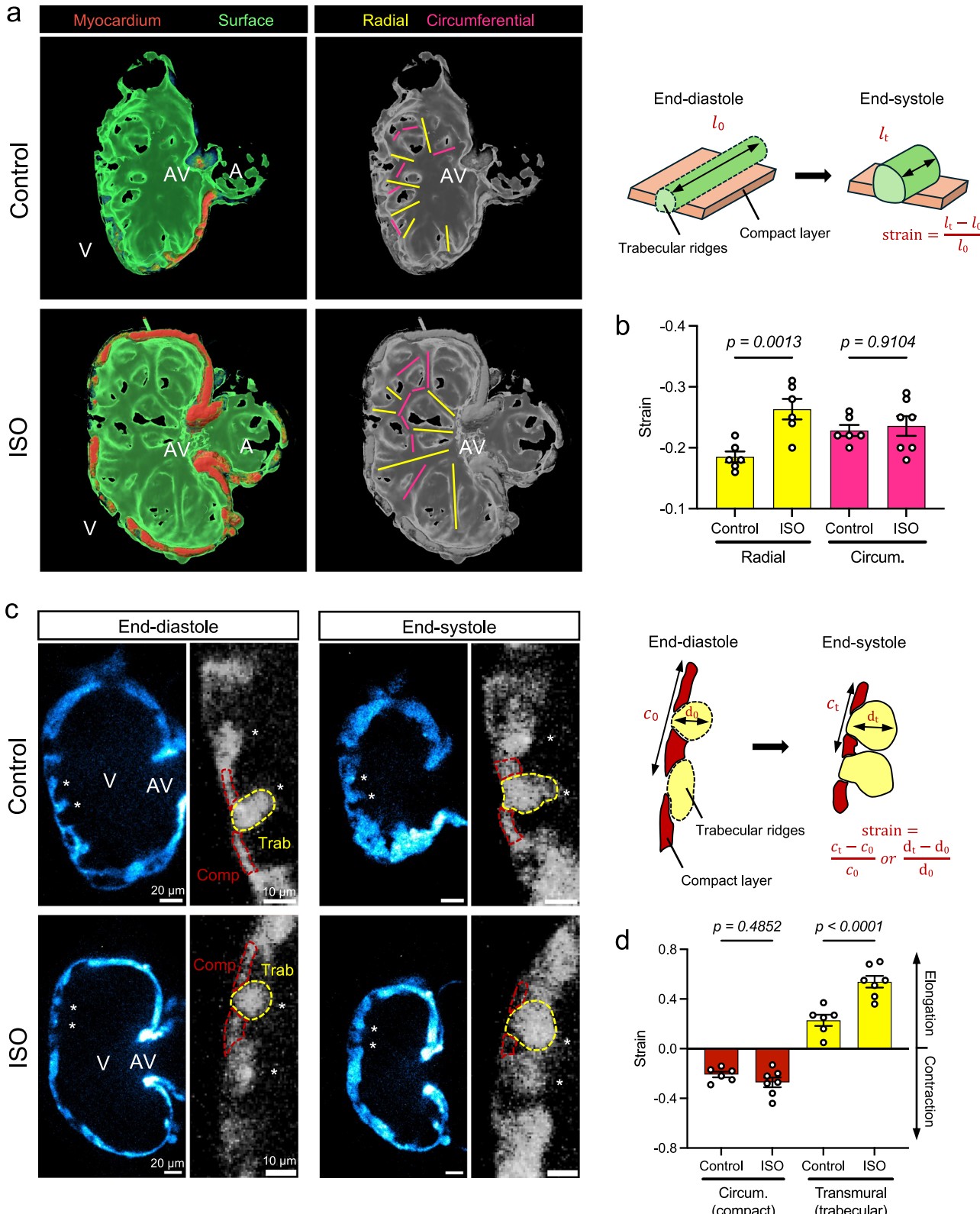

**Fig. 3 | ISO-mediated increase in strain aligning with radial trabecular ridges.**
**a**, **b** 3-D volumetric and surface rendering revealed the network of radial (yellow) and circumferential (pink) trabecular ridges inside the hearts at 4 dpf. ISO treatment significantly increased the strain (i.e., shortening) along the radial trabecular ridges (n = 6 ridges for each group) but not the circumferential ridges (n = 7 ridges for control, n = 7 for ISO). **c**, **d** 2-D cross-sections of the trabeculae (asterisks) revealed that the surrounding compact myocardium (red dashed outlines) contracts circumferentially while the ridges (yellow dashed outlines) thicken (elongate) transmurally during systole. ISO treatment significantly increased the transmural strain within trabecular ridges (n = 6 ridges for control, n = 7 for ISO), whereas the circumferential strain in the surrounding compact layer remained similar (n = 6 ridges for control, n = 7 for ISO). Anatomic labels: V ventricle, A atrium, AV AV canal. All values are displayed with mean and standard error of mean (SEM). *p*-value is displayed for each comparison. One control heart and one ISO-treated heart were used for analysis. Ordinary one-way ANOVA followed by Šídák's multiple comparisons test on the means was applied to determine statistical significance. Source data are provided as a Source Data file.

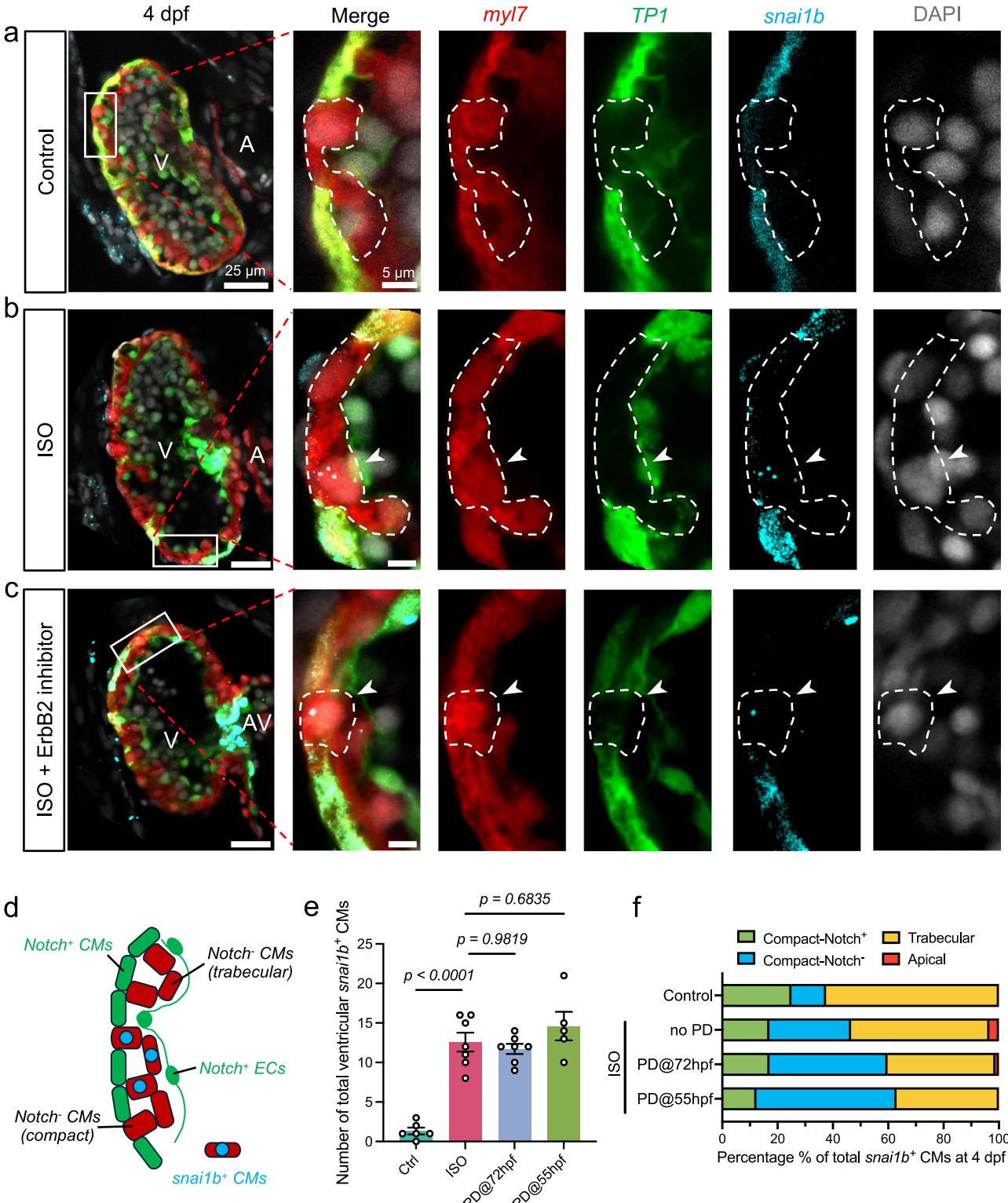

and subsequent EMT[44]. TGF-β binds to the receptor complex formed by TGFBR1 (type I receptor) and TGFBR2 (type II receptor), which phosphorylates SMAD transcription factors to turn on downstream genes[45]. We treated the embryos with LY364947, a selective inhibitor of TGFBR1, from 3 to 5 dpf to abrogate the TGF-β signaling[46]. The treatment significantly impaired trabeculation but not the myocardial *snai1b* expression (Fig. 7e–h). Taken together, our data suggested that *snai1b*-positive CMs do not share the canonical pathways with those of *snai1b*-positive epicardial and valvular cells. Consistent with the previous report, the tissue-specific *snai1b* may bypass the master

regulators (e.g., TGF-β, ErbB2) that control other Snail genes, such as *twist1b and snai2 (slug)*[21].

## Discussion

Biomechanical signals orchestrate the intricate coordination and organization of cardiac morphogenesis, and the mechanosensitive transcription factors mediate the initiation of trabeculation[47]. Snail-family transcription factors respond to hemodynamic shear stress to initiate endothelial-mesenchymal transition (EndoMT) during valvulogenesis[22,23]. Despite a paucity of data about Snail genes in

**Fig. 4 | _snai1b_ activation in Notch-negative cardiomyocytes undergoing delamination and trabeculation. a** At 4 dpf, myocardial Notch signaling (_TP1_) develops at the outer curvature of compact (cortical) layer, and discontinuation was observed when trabeculation sprouting occurs (dashed outlines in magnified sub-panels). Endocardial Notch mediates the ErbB2 signaling in trabecular cardiomyocytes (CMs) to activate Notch in the adjacent CMs and inhibit their delamination. On average, 1.3 _snai1b_+ CMs were found per ventricle (see (**e**), n = 6). **b** After ISO treatment, the gap between Notch-positive (+) CMs widened, and _snai1b_ activation (arrowheads, via in situ hybridization) was observed in the Notch-negative (−) CMs in the compact layer (29.5%) and trabeculation (50%), and a small number of apically delaminated CMs (3.4%) (see Fig. S3a–c). On average, 12.6 _snai1b_+ CMs were found per ventricle (see **e, f**, n = 7). **c** Co-treatment with ErbB2 inhibitor, PD168393,

reduced trabeculation (see Fig. S3d) but did not inhibit _snai1b_ activation (arrowheads, see **e, f**). **d** Schematic summarizing the Notch and _snai1b_ expression pattern in (**a**–**c**). **e, f** At 4 dpf, ISO treatment led to a 10-fold increase in the total number of _snai1b_+ CMs per ventricle. ErbB2 inhibitor (PD) co-treatments at 55 hpf (n = 5) or 72 hpf (n = 7) did not change the total number of _snai1b_+ CMs. Rather, they reduced the total trabecular population from 50 to 37% and 39%, respectively. All values in panel e are displayed with mean and standard error of mean (SEM). _p_-value is displayed for each comparison. Ordinary one-way ANOVA followed by Šídák's multiple comparisons test on the means was applied to determine statistical significance. Source data are provided as a Source Data file. Anatomic labels: V ventricle, A atrium, AV atrioventricular canal.

cardiomyocytes, myosin contraction was reported to activate Snail-dependent EMT during gastrulation[48]. Employing the myocardial Tol2-CRISPR activation/interference (CRISPRa/i) system, 4-D strain analysis, and single-cell transcriptomics, we demonstrate the unique _snai1b_+/Notch− cardiomyocytes responsive to ISO-mediated increase in myocardial strain for delamination and trabeculation.

The spatial distribution of _snai1b_ aligns with local spatial variation in strain and Notch-mediated trabecular-compact CM differentiation[4,13]. Earlier studies revealed that _snai1b_ expression was unchanged in the AV valves of mutants with reduced hemodynamic shear stress (_gata1_−/−) or shear stress-sensitive mechanotransduction (_nfatc1_−/−)[22,23]. As a corollary, at 54 hpf, we inserted 50-μm silica microbeads into the heart to increase endocardial shear stress (Supplementary Fig. 4d)[49], and at 5 dpf, no _snai1b_+ CMs were observed in the ventricle (sham n = 6, bead-inserted n = 9) (Supplementary Fig. 4e). Together with our finding using ErbB2 inhibitors, activation of myocardial _snai1b_ is independent of hemodynamic shear stress.

Instead, _snai1b_ is activated by an increase in myocardial strain within trabecular CMs. Both mRNA staining and transgenic reporter presented low _snai1b_ expression under physiological contractility. The scRNA-seq dataset was also consistent with the sparsity of _snai1b_+ CMs in vivo. ISO treatment preferentially increased the strain along radial trabeculae rather than the compact layer. The differential strain between the trabecular and compact layers is supported by their distinct cell morphology and myofibrillar architecture[37,50]. Compact CMs are disk-like and organize their myofilaments in a network fashion, whereas trabecular CMs are tubular, with myofilaments running densely in parallel to their cell body. Compared to compact CMs, the trabecular CMs develop a more mature force-generating phenotype and are likely more sensitive to strain perturbation by ISO or myosin inhibitors. This distinction of mechanical force may have determined the spatial distribution of _snai1b-positive_ trabecular CMs. In line with some recent evidence, Notch activation in compact CMs attenuates their actomyosin tension, which reinforces the local strain variation underlying _snai1b_+ CM-initiated delamination[51]. Our current study performed 2D deformation analysis on trabecular and compact CMs due to the difficulty of obtaining precise borders of the cells. In the future, it would be plausible to optimize the imaging and segmentation pipeline for 3D tissue strain analysis within the individual CMs.

_snai1b_ ensures the spatially organized delamination of trabecular CMs. In the presence of ISO-mediated strain, _snai1b_-repressed CMs remained in the compact layer or underwent apical delamination, whereas _snai1b_ activation initiated CM delamination and trabeculation. Other studies reported that _snai1b_ modulates the migration of myocardial progenitors before 24 hpf[26] and the organization of cytoskeleton from 48 to 52 hpf[27]. Here, our CRISPRa/i data extended to 96 hpf, demonstrating that _snai1b_ activation organizes cardiac trabeculation.

Myocardial _snai1b_ seems to obviate some canonical aspects of EMT, possibly due to tissue-specific enhancer activity. Our scRNA-seq analysis consistently predicted the expression of _col1a2_ within epicardial and valvular _snai1b_+ cells. Myocardial _snai1b_ did not induce

_col1a2_ expression and was not regulated by TGF-β or ErbB2 signaling, suggesting a unique mesenchymal-like transcriptome of _snai1b_+ CMs. In one study, _snai1b_ was the sole Snail gene that was not downregulated by ErbB2 inhibitors in the zebrafish heart[14]. In another study, TGF-β inhibition reduced _twist1b_ expression and the number of Snail+ CMs during cardiac regeneration; however, the Snail antibody was designed to recognize both Snai1 and Snai2 (Slug)[21]. TGF-β inhibition leads to an absence of trabeculation in zebrafish, indicating a reduction in CM proliferation[52] and possibly other Snail genes underlying the delamination and migration of trabecular CMs. Nevertheless, the precise mechano-sensitive role of Snail genes in atrium vs. ventricle during cardiac development warrants further investigation[53].

Numerous genetic variants have been implicated in left ventricular non-compaction (LVNC) and explored via animal models[54]. Many involve sarcomeric genes, while others are mutations in common developmental pathways mentioned above, including Notch (_MIB1_)[55], ErbB2 (_NUM/NUMBL_)[56], and TGF-β (_PRDM16_)[57]. Despite Snail genes-mediated EMT for the formation of coronary vasculature, their expression in left ventricular non-compaction (LVNC) in human or murine myocardium has not been elucidated[58]. In this context, etiology of LVNC remains to be explored in the context of dynamical proliferation and maturation programming of the ventricular wall[59].

Overall, we employed a zebrafish model to uncover the initiation of trabecular organization. _snai1b_-positive/Notch-negative cardiomyocytes provide new mechanotransduction insights into delamination during ventricular development. The _snai1b_+/Notch− cardiomyocytes are responsive to radially increased myocardial strain, implicating spatially organized patterning during trabeculation by mechanical forces.

## Methods
### Zebrafish lines
All experiments with zebrafish were performed in compliance with and with the approval of a UCLA Institutional Animal Care and Use Committee protocol (ID: ARC-2015-055).

Adult zebrafish were raised and bred in the UCLA Zebrafish Core Facility according to standard protocols[60]. Embryos were cultured in E3 medium (5 mM NaCl, 0.17 mM KCl, 0.33 mM CaCl$_2$, 0.33 mM MgSO$_4$ in sterile diH$_2$O) at 28.5 °C for all the procedures. 0.003% (w/v) 1-phenyl-2-thiourea (PTU, Sigma) was added to the medium to suppress the pigmentation. Embryos were transferred to the core facility at 6 days post-fertilization (dpf) and raised to 14 dpf. Transgenic lines _Tg(myl7:mCherry)_ and _Tg(flk:mCherry)_ were provided by the UCLA zebrafish core. _Tg(snai1b:EGFP)_^zdl100Tg line was a kind gift from Rodney Stewart at the University of Utah[28]. _Tg(myl7:mCherry-zCdt1)_ line was generated from a FUCCI line kindly gifted by Kenneth Poss at Duke University[29]. _Tg(TP1:EGFP)_^um14Tg line was a kind gift from David Traver at UCSD and Nathan Lawson at the University of Massachusetts Medical School[38]. _Tg(myl7:mKate-CAAX)_ line was a kind gift from Deborah Yelon at UCSD. Sex was not considered in our study design as we focus on time points prior to sex differentiation in zebrafish (~45 dpf).

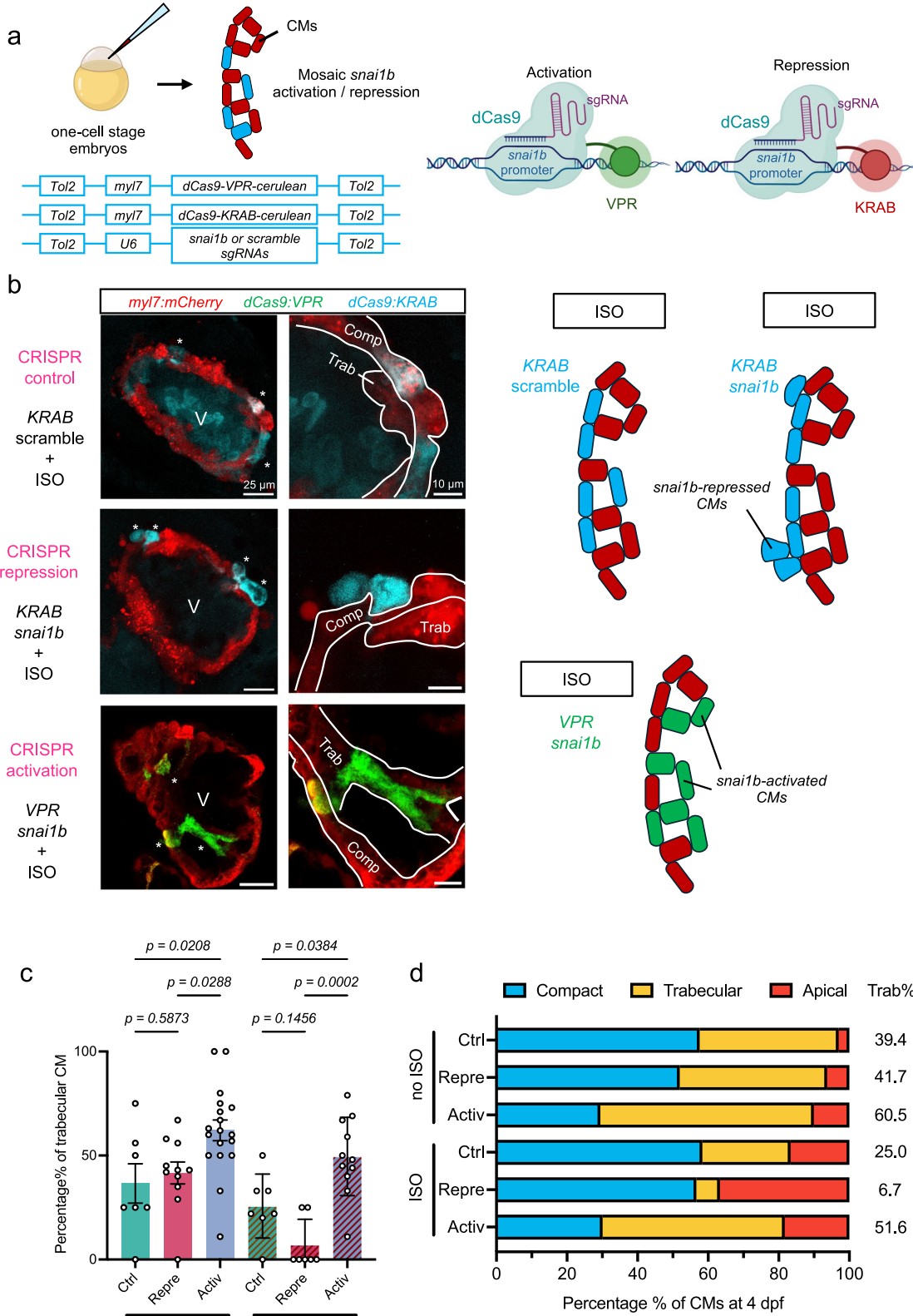

### Pharmacological treatment

Isoproterenol hydrochloride (ISO, Sigma, 100 μM) and 2,3-butanedione 2-monoxime (BDM, Sigma, 10 mM)[34] were dissolved in distilled water before being diluted in the E3 medium and applied to embryos at 24 h post-fertilization (hpf). Para-amino-blebbistatin (pAB, Cayman Chemical, 10 μM)[35], PD168393 (PD, Cayman Chemical, 5 μM)[4,14], and LY364947 (Abcam, 40 μM)[46] were first dissolved in DMSO before being diluted in E3. In control groups, pAB, PD, and LY364947 were replaced with DMSO at the same volume percentage. The solution is refreshed daily until imaging. For ISO-treatment past 5 dpf, the larvae were raised in core facility during the day (12 h) and kept in ISO solution at night (12 h) to ensure proper feeding.

### Microbead insertion

Zebrafish embryos were anesthetized in 0.2 g/L tricaine solution (Sigma) and mounted with yolk sac facing up in 1% low-melting point

**Fig. 5 | Activation and repression of *snai1b* modulates delamination for trabeculation. a** Experimental plan for the myocardial Tol2-CRISPR activation/interference (CRISPRa/i) system. Tol2 transposon plasmids were injected in one-cell stage embryos to induce mosaic activation or repression of *snai1b* among cardiomyocytes (CMs). In total, four sgRNAs that target different promoter regions of *snai1b* were used. Two scramble sgRNAs were used as control. Right panels show the mechanism of activation (VPR) and repression (KRAB). For system validation, see Supplementary Fig. 8. Partially created in BioRender. Wang, J. (2025) https://BioRender.com/ep5rh7k. **b**–**d** Under ISO treatment, the majority of control (58.3%) and *snai1b*-repressed (56.7%) CMs remained in the compact layer at 4 dpf (96 hpf). Activation of *snai1b* led to significantly more delaminated (see Supplementary Fig. 9b) and trabecular CMs per heart (**c**), where 51.6% of total *snai1b*-activated CMs form trabeculae, compared to 25.0% of control and 6.7% of *snai1b*-repressed CMs (**d**). On the other hand, repression of *snai1b* resulted in 36.7% of CMs undergoing apical delamination. For representative images of CRISPRa/i-injected hearts without ISO, see Supplementary Fig. 9a. All values in (**c**) are displayed with mean and standard error of mean (SEM). *p*-value is displayed for each comparison. Number of hearts analyzed: Control-ISO = 7, Repression-ISO = 7, Activation-ISO = 11. Ordinary one-way ANOVA followed by Holm–Šídák's multiple comparisons test on the means was applied to determine statistical significance. Source data are provided as a Source Data file. Anatomic labels: V ventricle, Comp compact layer, Trab trabeculae.

agarose (Thermo Fisher). A small opening was created in the yolk sac using a fine needle, through which the microbeads (50 μm in diameter, Alpha Nanotech) were delivered to the venous pole and, subsequently, the heart[49]. Afterward, the embryos were released from the agarose and returned to E3 medium at 28.5 °C until imaging.

## Confocal imaging
For whole-mount imaging, zebrafish embryos were fixed in 4% paraformaldehyde (PFA) solution overnight at 4 °C and washed in 0.1% PBST solution (Tween 20 in PBS). For whole-heart imaging, the fish were euthanized at 14 dpf by rapid cooling and fixed in 4% PFA overnight. After PBST washing, hearts were then dissected before imaging. If nuclei counterstain was applied, the samples were incubated with DAPI (Sigma, 1 μg/mL) in PBST for 4 h. The hearts and embryos were mounted in 1% agarose on No.1 cover glass (thickness ~0.17 mm) for imaging.

Imaging was performed on a Leica SP8 scanning laser confocal microscope using the control software LAS X (ver 3.5.7) from the Advanced Light Microscopy and Spectroscopy Lab at UCLA (RRID: SCR_022789). A 20X (NA = 0.75, #506343) or ×63 (NA = 1.20, #506346) water-immersion lens was installed for the imaging session.

## Whole-mount fluorescence in situ hybridization (FISH), imaging and quantification
We modified an established FISH-antibody staining protocol[61] according to the manufacturer's recommendations for the RNAscope technology (Advanced Cell Diagnostics). Briefly, the embryos were fixed in 4% PFA overnight at 4 °C and washed in 0.1% PBST. The skin around the heart was removed using a pair of fine forceps. The embryos were treated with 100% methanol for at least 2 h before being transferred to 3% (vol/vol) H₂O₂/methanol. After an hour of incubation, samples were rehydrated serially from 75% methanol (in 0.1% PBST) to 0.1% PBST. Tissue permeabilization was performed using first 1% Triton-X (in PBS, 1 h) and then RNAscope Protease Plus solution (30 min, 40 °C). Next, RNAscope probes were added to the samples, and the FISH signal was developed step by step following the user manual for the RNAscope Multiplex Fluorescent v2 Assay. Opal 690 and Opal 520 (Akoya Biosciences, 1:1000) were used to develop the fluorescent signal for RNAscope probes, including the negative control probes. After FISH, anti-GFP (GeneTex GTX113617) and anti-mCherry (Invitrogen M11217) primary antibodies (1:100) were incubated with the samples in the co-detection antibody diluent (Advanced Cell Diagnostics) overnight at 4 °C. Alexa Fluor 488 and Alexa Fluor 594 secondary antibodies (Invitrogen A-11008 and A-11007, 1:500) were then added along with DAPI to develop the signals for *snai1b/TP1*, *myl7*, and nuclei counterstain, respectively. The z-stacks of stained hearts were acquired using a Leica SP8 confocal microscope, as described above. For MF20 staining, anti-GFP and anti-mCherry was used to amplify the signal of *dCas9-EGFP* and *myl7*. Anti-MF20 antibody (Invitrogen 14-6503-82) was diluted at 1:500, and Alexa Fluor 647 (Invitrogen A-21235, 1:500) was used as its secondary antibody. The probes and antibodies used in this study can be found in the Supplementary Table 1.

## Tol2-CRISPR activation/interference
The *myl7(cmlc2)-dCas9-KRAB/VPR-cerulean* and *U6-sgRNAs/scramble* plasmids were obtained from VectorBuilder (Chicago, USA). *snai1b* sgRNAs were selected using CHOPCHOP[62]. Searches for CRISPR activation and repression resulted in identical top 4 target sequences (Genome version: GRCz11), which were then cloned into two sgRNA plasmids. Research has shown that at least 4 sgRNAs are needed for effective CRISPR activation and repression[39]. The scramble sgRNAs originated from the Alt-R® CRISPR-Cas9 Negative Control kit (Integrated DNA Technologies, Coralville, USA)[63]. IDs for all vector designs can be found in the Supplementary Table 1, and the vector maps are included in the supplemental information. The final injection mix contains: 50 ng/μL Tol2 transposase RNA, 10 ng/μL Tol2-dCas9-KRAB/VPR plasmid DNA, 10 ng/μL U6-sgRNA#1-2 plasmid DNA, 10 ng/μL U6-sgRNA#3-4 plasmid DNA, and 0.25% (w/v) phenol red[64]. For the control group, 20 ng/μL U6-scramble#1-2 plasmid DNA replaces the sgRNA plasmid DNAs. Microinjection was performed using a U-PUMP microinjector (World Precision Instruments, Sarasota, USA) at one-cell stage. Each embryo received 1 nL of injection mix and was returned to the incubator until screening for cerulean fluorescence at 3 dpf. If indicated, ISO treatment was applied to the injected embryos at 1 dpf. For the following groups, *dCas9-EGFP* plasmids were injected to replace the *dCas9-cerulean* plasmids: VPR-ISO, KRAB no ISO, control no ISO. All plasmids used in this study can be found in VectorBuilder database. The vector IDs are provided in Supplementary Table 1.

## Confocal image processing and quantification
To quantify the *snai1b:EGFP* fluorescence intensity, maximum z-projection was first performed on the confocal z-stacks using FIJI[65]. Then, ROIs were drawn around the region of interest (ventricle, BV annulus, bulbus), and the average intensity per pixel was measured. The raw EGFP intensity measurements were also normalized across embryos using the mCherry intensity as a scaling reference. Counting of *snai1b*⁺ cells was performed manually in FIJI or Leica LAS-X software.

## 4D spinning-disk fluorescence imaging
A spinning-disk confocal microscope was used to capture 3D+time images of the ventricular myocardium (scanning unit: Yokogawa CSU-X1; microscope: Leica DMI 6000B; camera: Andor iXon plus 897; objectives: Leica HC PL IRAPO 40× or 20× water immersion lens; control software: Micro-Manager 2.0 with custom control code). Live *Tg(snai1b:EGFP; myl7:mCherry)* embryos with (n = 3) or without (n = 3) isoproterenol treatment were anesthetized with tricaine as in confocal imaging and mounted in No.1.5 glass-bottom dishes (Mattek) with 1% agarose prior to imaging. At each z-position, 300 frames were acquired at 30 Hz (frame size: 512 × 512 pixels). For trabecular strain analysis, we used an ORCA-Flash4.0 LT digital CMOS camera (Hamamatsu) and imaged at 200 Hz to improve the spatial resolution (n = 3 each for control and ISO groups). The z-axis step size was 1 μm. The resulting 4D stacks were synchronized using a custom-written MATLAB program and divided into 10 cardiac phases from systole to diastole per cycle[66].

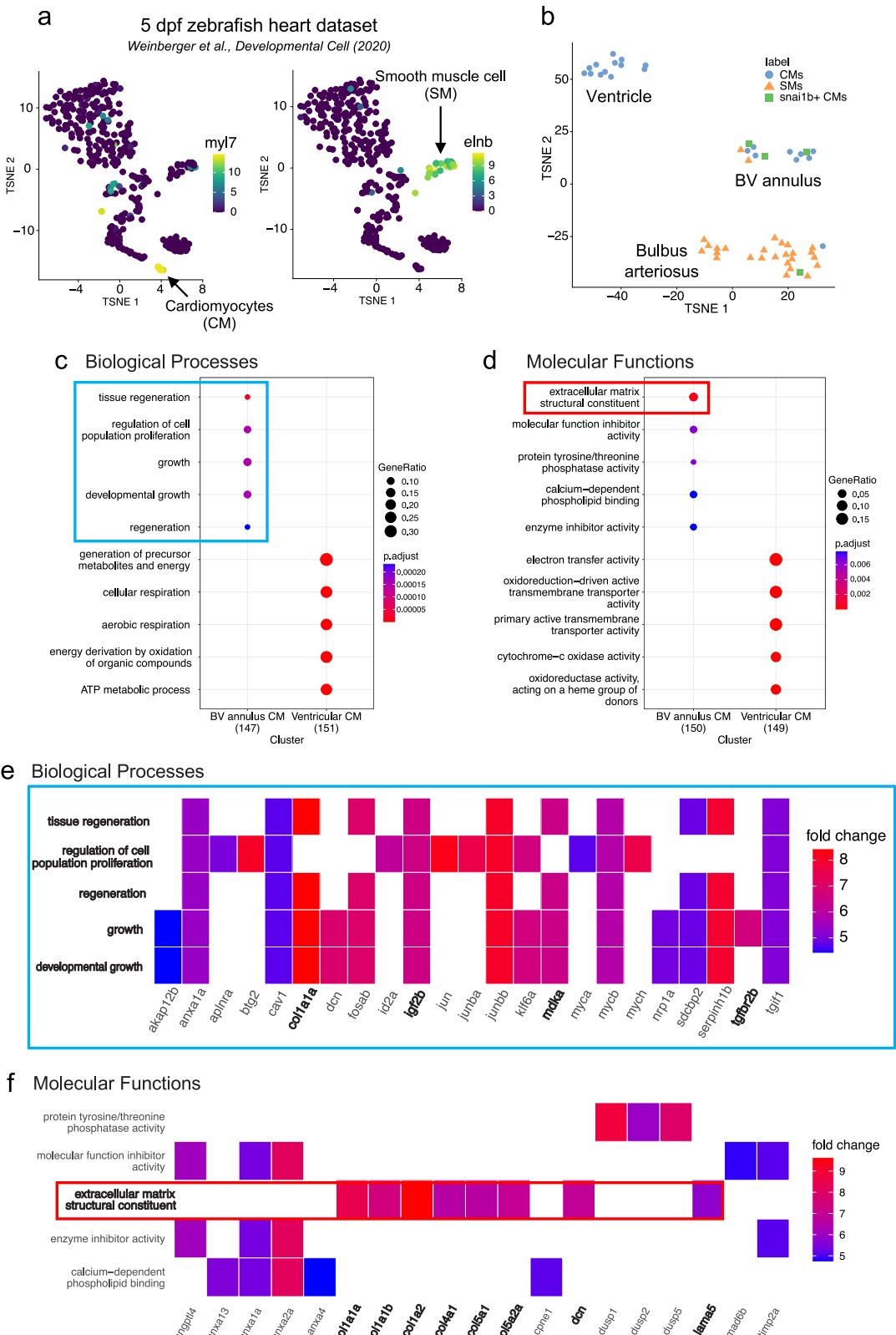

**4-D strain mapping**

Given the 3-D + time images for each ventricle, the following procedure was applied to quantify the area strain in the myocardium. First, basic image processing was performed using the Python libraries *SimpleITK*[67] and *scikit-image*[68], which included spatial resampling to reduce the computational burden and background removal using the rolling ball algorithm. The minimum-volume time points (i.e., cardiac phases) were identified manually and segmented in 3-D Slicer[69] using a combination of thresholding and manual editing. The segmented stacks were then converted to a triangulated surface mesh. Deformable image registration was conducted in a validated program that fits B-splines of Fourier to regularize the motion of images over one cardiac cycle and generates sequential deformations of the segmented myocardial surfaces[70]. The area strain for each triangular mesh

**Fig. 6 | Enrichment of mesenchymal genes in *snai1b*⁺ CMs. a, b** The CMs and SMs were isolated from a single-cell RNA seq dataset of 5-dpf hearts. These cells produced a sub-dataset with three clusters. *snai1b*⁺ CMs (green squares) were found in a cluster with a mixture of CMs and SMs, resembling the BV annulus region. The color scale indicates the normalized log counts of genes. The number of cells in the complete dataset was 366 after filtering, and the sub-dataset contains 53 cells. **c, d** Gene Ontology (GO) enrichment analysis categorized the enriched Biological Processes (BP) and Molecular Functions (MF). The BV annulus CMs had enriched mesenchymal BPs and MFs (blue and red boxes), compared to the ventricular CMs. The total number of marker genes selected for GO analysis was 151 for Ventricular

CM and 147 for BV annulus CMs. The dot size indicates the gene ratio (i.e., # of genes in a category divided by # of total genes). The color scale depicts the Benjamini & Hochberg (1995) adjusted *p*-values. The original *p*-values are calculated using a hypergeometric test (one-tailed) via the *enrichGO* function in clusterProfiler. Top five enriched terms with *p*-values < 0.05 from each group are displayed. Source data are provided as a Source Data file. **e, f** Heatmaps showing the genes involved in each of the BP and MF categories for BV annulus CMs. The color scale indicates the log-fold change in mean expression for each gene in BV annulus CMs compared to ventricular CMs. Top five enriched terms with *p*-values < 0.05 are displayed. Source data are provided as a Source Data file.

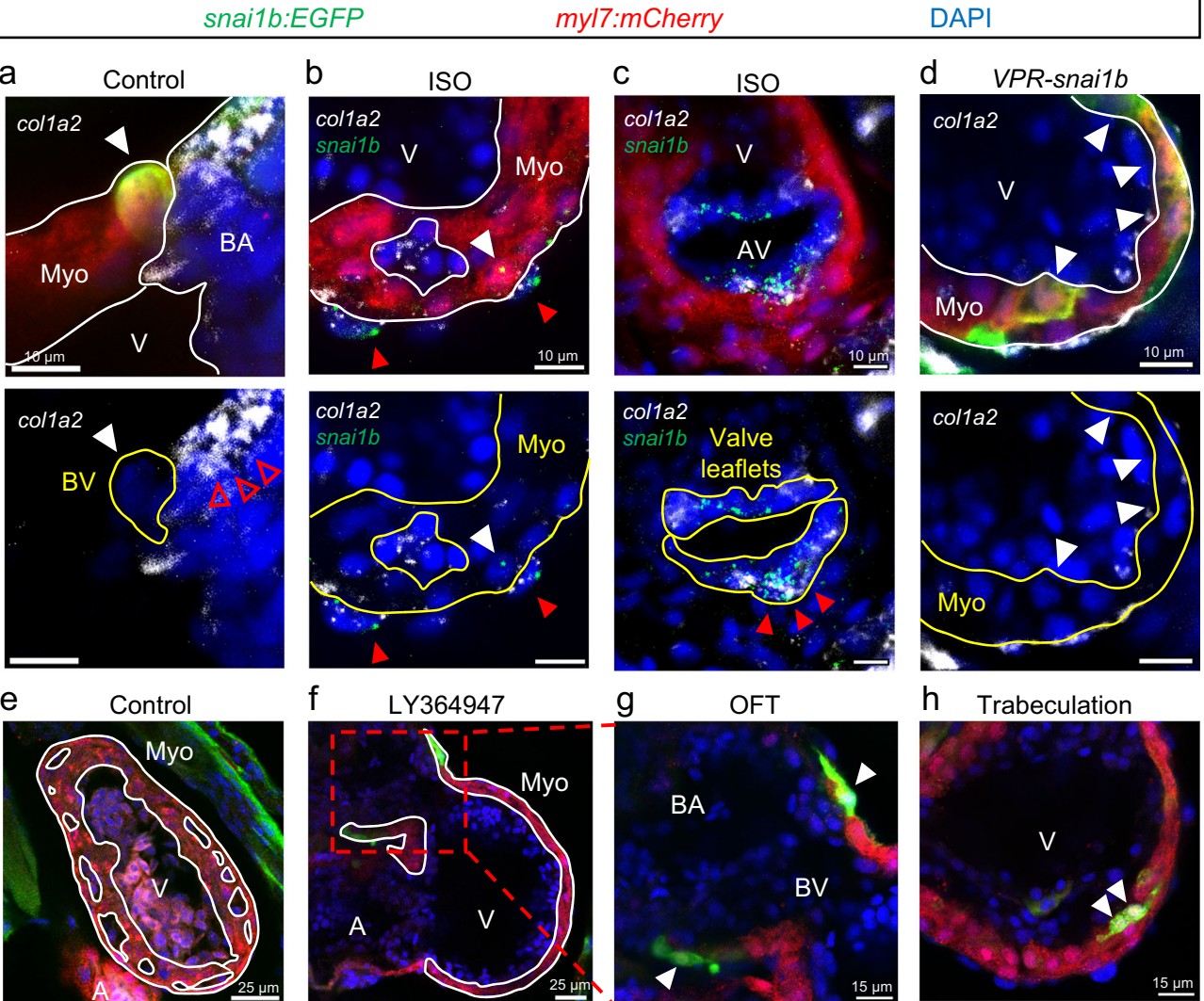

**Fig. 7 | *snai1b* pathways in CMs vs. epicardial and valvular cells. a–d** Abundant *col1a2* mRNA colocalized with *snai1b* mRNA in epicardial, valvular cells (red arrowheads), and bulbus smooth muscle cells (empty red arrowheads). However, *col1a2* expression was not observed in *snai1b*⁺ CMs (white arrowheads) of control (**a**), ISO-treated (**b**), and CRISPRa-injected (**d**) hearts at 4 dpf. Number of hearts: Control = 10, ISO = 6, CRISPRa (VPR) = 10. **e–h** At 3 dpf, LY364947, a TGF-β signaling

inhibitor, attenuated trabeculation (white outlines), but the expression of *snai1b* (arrowheads) within BV annulus and trabecular CMs persisted. Number of hearts: Control n = 10, LY364947 n = 12. BA bulbus arteriosus, V ventricle, BV bulbus-ventricular annulus, A atrium, Ch chest wall, Myo myocardium, AV atrioventricular canal.

element was computed as follows[32]:

$$\epsilon_{\text{area},j}(t) = \sqrt{\frac{A_j(t)}{A_j(0)}} \qquad (1)$$

Where $A_j(0)$ is the area of element $j$ at the initial state (minimum volume), and $A_j(t)$ is the area of element $j$ at subsequent states obtained by the registration and morphing process.

The computation results were outputted as VTK files and imported into ParaView (Kitware) for visualization. We selected a pair of control and ISO-treated samples that had the least artifacts from registration (e.g., unnatural displacements of mesh elements) for downstream analysis. The models were first warped around the displacement vectors (scale = −1) to remove the contractile motions. A clipping plane was applied to bisect the ventricles along the AV canal and the outflow tract. We selected the halves facing the objective lens for strain analysis. The

unselected halves suffered more from decreasing contrast as imaging planes approached the working distance of the objective lens. The inner (endocardial) and outer (epicardial) surfaces of the myocardium were separated using the Connectivity filter in ParaView. Triangular elements of both the epi- and endo-ventricular walls were then manually picked out using the polygon selection tool. Care was taken to ensure the elements from the two surfaces overlap. The area strain values were extracted to calculate the average strain at each time point. Strain variations within the sampling regions were calculated as the standard deviation of strains at each time point (phase).

For the trabecular strain analysis, the 3-D surface of trabecular network was rendered in Amira (Thermo Fisher), and the shortening of radial and circumferential trabecular ridges during systole was obtained using the 3-D Measurement tool. The transmural thickening (elongation) of trabecular ridges and the circumferential contraction of compact layer were measured in 2-D cross sections of trabeculae using FIJI. Finally, the data was delivered to GraphPad Prism for statistical analysis.

### Additional methods for cardiac functional measurement

To quantify the heart rate and ejection fraction of control and ISO-treated hearts, the same group of *Tg(myl7:mCherry)* embryos were imaged at each time point using an Olympus IX70 epi-fluorescent microscope (objective: UPLFLN 10X/0.3 NA) with a Prime BSI camera (exposure time: 10 ms, 2 × 2 binning, control software: Micro-Manager 1.4). Heart rate was extracted via kymograph in FIJI[71]. Ejection fraction was estimated by calculating: *(Diastolic ventricular area−Systolic ventricular area)/Diastolic ventricular area × 100%*[72]. Care was taken to ensure each embryo was imaged in the same orientation.

To quantify the changes in cross-sectional area of trabecular/compact CMs, *Tg(myl7:mKate-CAAX)* embryos were mounted in No.1.0 glass-bottom dishes (Mattek) with 1% agarose and imaged using a Leica SP8 digital light-sheet (DLS) system (camera: Hamamatsu Orca Flash 4.0 V2; illumination objective: Leica HC PL FLUOTAR 2.5×/0.07 Dry; detection objective: Leica HC PL FLUOTAR 25×/0.95 Water) at 200 frame per second. Measurements were performed in FIJI on slices near the center long-axis plane, where most of the cells were doing in-plane movements. Care was taken to measure only cells that appear in every frame of the time course.

### Single-cell RNA sequencing data analysis

The 5 dpf zebrafish heart dataset was downloaded from NCBI GEO (accession number GSE121750), where the post-QC matrix (M0) was used for our analysis[40]. The matrix was imported into a *SingleCellExperiment* (SCE) object in R[73]. Library normalization and dimensional reduction (PCA and TSNE) were performed using the R/Bioconductor package *scater*[74]. The cardiomyocytes (CMs) and smooth muscle cells (SMs) were found by thresholding the normalized log counts of their marker genes, *myl7* and *elnb/mylka*, respectively. Cells with marker log counts >2 were collected, and their unprocessed read counts in M0 were isolated to build a new matrix, followed by normalization and dimensional reduction. From this new matrix (M1), a t-SNE plot was generated to reveal the ventricle, the BV annulus, and the bulbus clusters.

For Gene Ontology (GO) analysis, the cardiomyocytes were first collected using *myl7* log counts >2 as the criteria from M1. These cardiomyocytes already carried the cluster labels (i.e., BV annulus and ventricle) and formed a new SCE object. Expression plots of scarcomeric genes were created from this SCE object via *scater*. Then, a list of top markers was generated for the BV annulus and the ventricular population by the *scran* package (log fold-change >2, $p < 0.05$) and passed down to the *clusterProfiler* package for enrichment analysis[75]. The GO dot plots and heatmaps were generated by the *enrichplot* package. The original *p*-values are calculated using a hypergeometric test (one-tailed) via the *enrichGO* function in clusterProfiler. Top five enriched terms with *p*-values < 0.05 from each group are displayed. In the dot plots, the color scale depicts *p*-values adjusted by the Benjamini-Hochberg (1995) method to control the false discovery rate.

### Statistics and reproducibility

All values are displayed with mean and standard error of mean (SEM). Ordinary one/two-way ANOVA followed by Šídák's (or Holm-Šídák's) multiple comparisons test on the means was applied to determine statistical significance. Paired t-test (two-tailed) was performed for Supplementary Fig. 3b, d. All statistical tests were performed in GraphPad Prism. A *p*-value below 0.05 was deemed statistically significant. The resulting *p*-values and the sample sizes (n) can be found in the corresponding figure legend. All data points for each statistical test were taken from distinct samples. Information of each statistical analysis can be found in the corresponding sheet of the Source Data file.

All experiments in this study were repeated independently at least twice with similar results (ISO treatment, ISO+ErbB2 inhibitor treatment, ISO+myosin inhibitor treatment, microbead insertion, TGF-β inhibition, and ISO + CRISPR activation/interference). No statistical methods were used to predetermine the sample size. Sample sizes were determined based on published studies by our group, availability of the embryos/larvae and feasibility required to confirm obtained results. Overall, at least 5 samples per group were obtained for each statistical comparison.

In CRISPRa/i experiments, samples with a single CM labeled were discarded as outliers. When obtaining data from whole-mount samples, hearts with severe distortion from the staining process were discarded to ensure accurate measurements. In the strain analysis, samples with severe motion artifacts post-synchronization were discarded. During experiments, embryos that died or showed severe cardiac dysfunction before reaching end points were discarded. No other exclusion was carried out. Embryos/larvae were randomly selected and distributed into experimental groups from large pool of spawns. During imaging analysis, data were distributed to more than one investigator who did not perform the experiment, and they were blinded from the experimental conditions.

### Reporting summary

Further information on research design is available in the Nature Portfolio Reporting Summary linked to this article.

## Data availability

The single-cell RNA seq dataset analyzed in this study can be downloaded from NCBI GEO (GSE121750), where the post-QC matrix (M0) was used[40]. The R data file for single-cell RNA seq analysis and the VTK files for strain mapping can be accessed via Zenodo repository at https://doi.org/10.5281/zenodo.10525228. Its DOI is also listed in Supplementary Table 1. Source data are provided with this paper. Any additional information required to reanalyze the data reported in this paper is available from the corresponding author upon request. Source data are provided with this paper.

## Code availability

The code for 4D imaging synchronization and myocardial strain mapping is custom-written and involve multiple software. Due to this non-turnkey nature, the code is not uploaded to public repository and will be readily shared upon requests. The original R script for scRNA-seq analysis can be accessed via Zenodo repository at https://doi.org/10.5281/zenodo.10525228. Its DOI and the software used in this study are listed in Supplementary Table 1. Any additional information required to reanalyze the data reported in this paper is available from the corresponding author upon request.

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

## Acknowledgements

We would like to thank the UCLA zebrafish core, Rodney Stewart (University of Utah), Kenneth Poss (Duke University), David Traver (UCSD), Deborah Yelon (UCSD), and Nathan Lawson (University of Massachusetts Medical School) for the fish lines. We especially appreciate Ching-Ling (Ellen) Lien (Children's Hospital Los Angeles), Julien Vermot (Imperial College London), and Hajime Fukui (Tokushima University) for advising the project. We are also grateful for the microscopy expertise from the Advanced Light Microscopy and Spectroscopy Lab at UCLA (RRID: SCR_022789). This study is funded by NIH grants R01HL129727 (A.L.M., T.K.H.), R01HL159970 (A.L.M., T.K.H.), R01HL165318 (T.K.H.), and T32HL144449 (E.Z., T.K.H.).

## Author contributions

Conceptualization, J.W. and T.K.H.; Methodology, J.W., A.L.B., S.-K.P., A.L., E.Z., R.O., P.Z., J.J.H.; Investigation, J.W., A.L.B., S.-K.P., C.Z.Z., R.O., and P.Z.; Writing— original draft, J.W., A.L.B., and T.K.H.; Writing—review and editing, J.W., A.L.B, S.-K.P., C.Z.Z., A.L., E.Z., T.Y., J.L., J.-N.C., A.L.M., and T.K.H.; Funding acquisition, A.L.M. and T.K.H.; Supervision, A.L.M. and T.K.H.

## Competing interests

The authors declare no competing interests.
