## [Transparent Peer Review file · Nature Communications]

Mechanically Activated *snai1b* Coordinates the Initiation of Myocardial Delamination for Trabeculation

Corresponding Author: Dr Tzung Hsiai

Version 0:

Reviewer comments:

Reviewer #1

(Remarks to the Author)

1. Summary:

This study reports the presence of *snai1b* expressing cardiomyocytes in the developing zebrafish heart. The authors observed an increase in the number of *snai1b*⁺ CMs at 4dpf upon stressing the heart with Isoproterenol (ISO) treatment. They also reported that ISO treatment generated higher strain variation in the ventricle, mainly contributed by the trabecular ridges. From these observations and current literature, they hypothesize that the strain variation from the treatment is giving rise to *snai1b*⁺ delaminating CMs. They also reported an increase in the percentage of notch negative compact layer CMs expressing *snai1b* with ISO treatment at 4 dpf. For further validation, the authors reported that upon CRISPR activation of *snai1b* in CMs, they saw increased contribution to the trabecular layer in the absence of ISO, and upon CRISPR-repression of *snai1b*, reduced trabecular contribution with ISO treatment. Single-cell RNA sequencing of 5dpf hearts revealed that *snai1b*⁺ CMs were lacking EMT-related ECM genes, hence the authors tested the role of TGFβ which is a known regulator of EMT. They concluded that *snai1b*⁺ CMs may be regulated by other non-canonical signalling pathways.

2. Major concerns/ suggestions

- a. In Figure 1-2, the expression pattern of *snai1b* remains unclear during the course of initiation of trabeculation and further expansion in untreated hearts. A time tracking experiment with a *snai1b*-EGFP line during the window of 54 hpf till 120 hpf should help establish this key point.
- b. In all experiments from Figure 2, ISO was administered at 1 dpf and onwards, and in Supplementary Figure 2, co-treatments with ISO and pAB or BDM were also performed from 1 dpf which is an early time point, and might affect the overall cardiac development leading to secondary effects. Have the authors tried to start treatments at a later time point to observe changes in the frequency of *snai1b*⁺ CMs?
- c. The *snai1b*⁺ CMs are not clearly visible in the control in Figure 2, panel (a).
- d. In Figure 2, panel (c), it would make the Figure more impactful if we could see the *snai1b*-EGFP⁺ CM contribution to the trabecular layer in untreated 14 dpf hearts.
- e. In Figure 3, panel (a), the representative images for the ventricle show a difference in chamber size. In panel (c), the representative image for control heart appears to have more complex trabeculation than the ISO treated. Are these observations consistent in all ISO treated hearts? A quantification of ventricular chamber size, total CM number, and trabecular CM number will help clarify this.
- f. The hypothesis that *snai1b* is expressed in delaminating CMs as a result of strain variation/tension heterogeneity needs to also be verified during the physiological delamination process due to tension heterogeneity between 54-65 hpf window.
- g. The authors claim an average of 1.3 *snai1b*⁺ CM in a control 4 dpf heart, but Figure 4, panel (a) is not representative of the plot in Figure 4, panel (e).
- h. In Figure 4, using a membrane line for marking the cardiomyocytes will help with getting resolution on marking and counting compact, delaminating, and trabecular CMs.
- i. In Figure 5, the authors intend to look at number of CMs undergoing delamination to form trabeculae upon activation/repression of *snai1b*. In physiological conditions, the most active window for observing delamination lies prior to 4dpf. An experiment checking for earlier time points will help with acquiring precise number of changes in delaminating CMs.
- j. The authors claim that *snai1b*⁺ CMs are not regulated by canonical EMT pathways, but do not provide any data about the actual regulatory mechanism.

(Remarks on code availability)

Reviewer #2

(Remarks to the Author)

This research explores the role of the *snai1b* gene in the early stages of heart development, specifically in the process of trabeculation, where the heart muscle forms ridges essential for heart function.

Mechanics Comments

1. For the strain quantification in Figure 2d-g:

a. Please clarify what type of images were used for strain analysis analyzed, so as to verify that strains are indeed analyzed at the myocardial, instead of, say, the endocardium.

b. Will quantifying strains for the inner surface alone be misleading? ideally, the mid-wall surface, cutting right through the myocardium, rather than on either the epicardial or luminal side of the ventricular wall. This is so that we can be sure that strain measurements are done within the target tissue. Further, the inner surface of the heart will be very undulating with trabeculation structure, and strains can be hard to extract, especially when neighbouring trabecular structure touch during systole and become fused together in the reconstruction, causing strain errors. How is this managed? From this aspect, it seems that the inner surface may have more strain artefacts than the outer, smoother surface.

c. A worrying sign of high errors in the strain quantification is that in Fig 2g, the strain variations (explained as max areal strain – min areal strain) does not go down towards zero at the end of the cardiac cycle, when the heart should have returned to its original shape. As Fig 2g is used as a primary evidence for increased spatial variability of strains, this issue may impact the credibility of the claim. It would be good to see the authors' validation of the strain quantification codes.

d. Perhaps instead of claiming spatial variability – which I think may be very difficult, the authors should tone down to claim that ISO hearts generally contract and deform more, and leave the link between a greater deformation and greater spatial variability as an unproved and logical speculation.

2. The division of the trabeculation into radial and circumferential may be contentious, some of the yellow lines in figure 3 do not seem to point to the AVC but are longitudinal in nature (apex – outflow tract in orientation, vertical in the images), but they are considered to be radial, one of the pink lines seem to be very close to the “left-right” axis, but is not considered radial. What is the specific definition of radial and what is the deviation tolerance of the definition? A concern is that this may be an unnatural pick-and-choose approach to generate statistical significance.

3. The trabeculation will naturally deform more than compact layer, as rightly pointed out by authors, this is due to myofiber alignment, but it may also be because the trabeculation has room to thicken, while compact layer is crowded and does not. My questions are:

a. The authors claim that because trabecular thickening is statistically significant with and without ISO while compact layer shortening is not, this means that there is higher spatial strain variation with ISO – this is not easy to accept. It is easy to imagine many scenarios where the observation holds true but spatial variation is not increased. For example, strains are proportionally increased with ISO in the same way everywhere, but measurement errors cause the compact layer data to be statistically insignificant due to its lower magnitudes. The authors should directly quantify spatial variation in strain if they wish to make this claim.

b. It is also not clear why the difference between the trabecular and compact layer strain is the main point here. To be relevant to trabeculation formation and delamination, we really need to look for localized variation in strain between neighbouring cells from the same layer in the pre-trabeculated myocardial wall. After the trabeculation is formed, there will naturally be differences in strains between trabeculation and compact layers, and it seems that this trabecular-compact strain difference will not contribute to any further trabecular tissue formation. It seems that the wrong type of spatial strain variation is used to substantiate a claim that there is a pro-trabeculation biomechanical environment.

Other Thoughts:

1. If *snai1b*+ CM is so sparse, only 1-2 cells per heart, this does not seem enough to support the large areas of trabeculation occurring in the heart. Does this mean that *snai1b* is not important to natural trabeculation?

2. It seems good to have a basic demonstration of which condition will result in more trabeculation and which will have less, via plain microscopy imaging, and quantification of trabeculation tissue area/volume. This is not so easy to appreciate from images with very busy fluorescent labels (eg., fig 5).

3. The authors did not show a comparison of *snai1b*+ with wildtype without ISO, it seems that such a baseline control will be important.

Biological comments:

1. The quantification of snai1 activation (Fig.1a-b) is confusing. The authors should include the number of cells positive along with the number of animals analysed. They should also quantify where the ectopic cells activating Snail are located.
2. Isoproterenol treatments: the treatment starts at 1 dpf and heart are analysed at 4dpf or 14dpf. The authors never show if the treatment is efficient in activating strain throughout the length of the treatment. I would think that the effect of the drug will vanish with time. A possibility is that the early treatment leads to abnormal development of the heart leading to aberrant force distribution in the contracting heart, thus independent from the change in strain. The authors need to clarify this.
3. The cell deformation in response to strain described in figure 3 is not conclusive. How can you conclude without having a three dimensional analysis of the single cell? The change in shape could simply be due to an artefact due to a different plane of view.
4. Fig 6 and 7 are not completely relevant to the study, not sure they belong to main figures

(Remarks on code availability)

Reviewer #3

(Remarks to the Author)

This study explores the role of snai1b-positive cardiomyocytes (CMs) in the initiation of trabeculation during heart development in zebrafish. Unlike typical Snail family genes involved in epithelial-to-mesenchymal transition (EMT), these snai1b+ CMs are mechano-sensitive and initiate delamination for trabecular organization. At 4 days post-fertilization, snai1b+ CMs were sparse, but their numbers increased significantly when myocardial strain was enhanced using Isoproterenol (ISO). Most of these CMs were Notch-negative. CRISPR-activation of snai1b promoted trabeculation, while its repression reduced trabecular formation, even under strain. The study also found that snai1b+ CMs do not exhibit typical EMT markers, like collagen 1a2 production. These findings suggest that snai1b+ Notch-negative CMs are mechanically activated to drive cardiac trabeculation independently of traditional EMT pathways.

In this article, the authors describe how strain-induced snai1b expression in notch1-repressed cardiomyocytes elicits their delamination, the initial step of trabeculation. They demonstrate the increase in snai1b+ CM numbers using a reporter of snai1b that does not seem to recapitulate snai1b expression, neither in the epicardium and the endocardium, as expressed by the authors in lines 92-94, nor in the cardiomyocytes, contrary to their claim in those lines, as seen in Figure S1 and Fig4a. There, expression of snai1b is widespread in almost all cardiomyocytes. Thus, validity of this reporter in the context of the heart is not, to my view, sufficiently proved.

However, the RNA-seq reanalysis performed was a fruitful approach and the subsequent experiments on snai1b-mediated regulation of col1a2 and TGF- β induction of snai1b expression, showing independency of this signals in the regulation of the delamination of CMs is surprising but very interesting.

Several experiments would be needed, specifically:

-To further validate the reporter in this system: mRNA expression analysis (as in Supp. Fig. 1 or in Fig. 4a) at different stages to compare with reporter data.

-To supplement the data in figure 4, maybe the opposite experiment, using Notch1 ICD IHC to show absence of colocalization with snai1b+ CMs, as data shown in Fig. 4 a-c do not correspond to quantification in e.

-Figure 4, it would be useful to quantify the Notch+ CMs, as they seem to decrease in number when ErbB2 is inhibited.

-In line 207, related to data shown in Figure 4, the authors show a decrease in trabeculation when ErbB2 is inhibited. But there is no quantitative data and panels shown in Suppl. Fig. 3 are not that clear. Can they provide these quantitative data?

-In figure 5 an appropriate explanation of why the snai1b-activated fish were not subject to ISO treatment is necessary, as well as the comparison with snai1b repressed without isoproterenol.

-Fig. 5c refers to percentage of delaminating CMs/cerulean + in KRAB? Which normalization is then used for VPR?

Otherwise, is it divided by the total number of CMs?

-The claim that snai1b repressed cells that "delaminated apically" are CMs is not properly substantiated, as they do not show any CM identity marker, besides the fact that myl7 is driving the expression of cerulean. But in suppl. Fig. 4c there are Cerulean+ cells that are not mCherry+, despite having the same transgenic strategy. In order to ensure that those cells are CMs, it would be interesting to have any cardiomyocyte marker analysed either by ISH or IHC.

-Regarding the scRNAseq data in figure 6, I think it would be informative to compare the snai1b+ CMs from the BV annulus to their snai1b- counterparts in the same tissue, as their contribution when comparing with ventricular CMs (which was the case, I think) might be minor (they represent only 27% of the CMs).

-Moreover, I think that, a scRNAseq experiment with ISO treated hearts can produce more useful information.

(Remarks on code availability)

Version 1:

Reviewer comments:

Reviewer #2

(Remarks to the Author)

The authors have addressed all the points, well done for this very interesting study.

(Remarks on code availability)

Reviewer #3

(Remarks to the Author)

This revised manuscript presents an insightful and well-executed study identifying a novel role for mechanically activated *snai1b*⁺ cardiomyocytes in initiating myocardial delamination and trabeculation in the developing zebrafish heart. The authors convincingly show that *snai1b* expression is induced by myocardial strain, independently of hemodynamic shear stress, and functions upstream of trabecular morphogenesis through a distinct, Notch-negative cell population. These findings represent a significant contribution to the understanding of cardiac mechanotransduction and ventricular wall patterning.

Strengths of the study include the combination of CRISPRa/i technology, 4D strain mapping, and single-cell/spatial transcriptomics, which together provide a robust and multidimensional view of *snai1b*'s role. The data are well-presented, and the experimental approach is methodologically sound. Importantly, the identification of *snai1b*⁺ CMs that diverge from canonical EMT and are selectively sensitive to mechanical cues advances the field beyond prior models reliant on shear stress or Notch signaling alone.

The revised version of this manuscript has improved it substantially and has answered my previous concerns and suggestions.

(Remarks on code availability)

Response to the reviewers' comments

Reviewer #1 (Remarks to the Author):

1. Summary:

This study reports the presence of *snai1b* expressing cardiomyocytes in the developing zebrafish heart. The authors observed an increase in the number of *snai1b*⁺ CMs at 4 dpf upon stressing the heart with Isoproterenol (ISO) treatment. They also reported that ISO treatment generated higher strain variation in the ventricle, mainly contributed by the trabecular ridges. From these observations and current literature, they hypothesize that the strain variation from the treatment is giving rise to *snai1b*⁺ delaminating CMs. They also reported an increase in the percentage of notch negative compact layer CMs expressing *snai1b* with ISO treatment at 4 dpf. For further validation, the authors reported that upon CRISPR activation of *snai1b* in CMs, they saw increased contribution to the trabecular layer in the absence of ISO, and upon CRISPR-repression of *snai1b*, reduced trabecular contribution with ISO treatment. Single-cell RNA sequencing of 5 dpf hearts revealed that *snai1b*⁺ CMs were lacking EMT-related ECM genes, hence the authors tested the role of TGF β which is a known regulator of EMT. They concluded that *snai1b*⁺ CMs may be regulated by other non-canonical signalling pathways.

2. Major concerns/ suggestions

a. In Figure 1-2, the expression pattern of *snai1b* remains unclear during the course of initiation of trabeculation and further expansion in untreated hearts. A time tracking experiment with a *snai1b*-EGFP line during the window of 54 hpf till 120 hpf should help establish this key point.

We thank the reviewer for their insightful suggestion. We have included data of *snai1b* expression from 56 hpf to 96 hpf, both reporter activity and mRNA staining, in the revised **Figure 1d-e** (see below) and **Supplementary Figure 2**. In untreated hearts, *snai1b* expression is predominantly around the BV annulus during trabeculation.

Figure 1.

(d) During trabeculation (56 hpf – 96 hpf), *snai1b:EGFP* signal is concentrated around the BV annulus. Isoproterenol (ISO) treatment at 1 dpf induced *snai1b* activation in the ventricular CMs.

(e) Whole-mount *in situ* hybridization of *snai1b* mRNA in the *Tg(snai1b:EGFP; myl7:mCherry)* reporter line reveals a significant increase of both EGFP+ and mRNA+ CMs in ISO-treated hearts. (see Supplementary Figure 1-2 for staining images). All values are displayed with mean and standard error of mean (SEM). *p*-value is displayed for each comparison. Number of hearts analyzed in each is displayed in panel d.

Anatomic labels: BA, bulbus arteriosus; V, ventricle; BV, bulbus-ventricular annulus; AV, atrioventricular canal.

Supplementary Figure 2. Whole-mount *in situ* hybridization of *snai1b* mRNA in the larval hearts of *Tg(snai1b:EGFP; myl7:mCherry)* reporter line. Related to Figure 1.

(a-d) Whole-mount *in situ* hybridization of *snai1b* mRNA (arrowheads) reveals that the reporter activity is myocardial-specific at 56 hpf **(a)**, 72 hpf **(b)**, 96 hpf **(c)**, and 5 dpf **(d)**.

(e) Percentage of *snai1b:EGFP*-positive cardiomyocytes (CMs) that are also positive for mRNA staining at 56 hpf (Ctrl n = 6, ISO n = 5), 72 hpf (Ctrl n = 7, ISO n = 7), and 96 hpf (Ctrl n = 6, ISO n = 6). All values are displayed with mean and standard error of mean (SEM). *p*-value is displayed for each comparison.

Anatomic labels: BA, bulbus arteriosus; V, ventricle; AV, atrioventricular canal; BV, bulbus-ventricular annulus; Myo, myocardium.

b. In all experiments from Figure 2, ISO was administered at 1 dpf and onwards, and in Supplementary Figure 2, co-treatments with ISO and pAB or BDM were also performed from 1 dpf which is an early time point, and might affect the overall cardiac development leading to secondary effects. Have the authors tried to start treatments at a later time point to observe changes in the frequency of *snai1b*⁺ CMs?

We thank the reviewer's suggestions. In the revised **Supplementary Figure 4c** (see below), we quantified the number of *snai1b*⁺ (both EGFP and mRNA) CMs at 96 hpf when treatments started at 72 hpf (3 dpf). Compared to the ISO-24 hpf group, we observed a significant decrease of EGFP⁺ but not mRNA⁺ CMs in the ISO-72 hpf group, as expected in mRNA transcription preceding EGFP translation. pAB co-treatment at 72 hpf significantly reduced the number of mRNA⁺ CMs, consistent with our 24 hpf data. Interestingly, BDM did not reduce the number of mRNA⁺ CMs. This may be due to insufficient dosage or off-target effects related to BDM (Hall *et al.*, 2016 – PMID: 26733241).

Supplementary Figure 4.

(c) ISO treatment starting at 72 hpf instead of 24 hpf reveals a significant decrease in EGFP⁺ CMs, but not mRNA⁺ CMs in the whole-mount *in situ* hybridization of *snai1b* mRNA in *Tg(snai1b:EGFP; myl7:mCherry)* hearts at 96 hpf. ISO at 72 hpf plus pAB treatment led to a significant decrease in mRNA⁺ CMs ($p = 0.0117$ vs. ISO alone). All values are displayed with mean and standard error of mean (SEM). p -value is displayed for each comparison. Number of hearts analyzed: ISO-24hpf = 6, ISO-72hpf = 7, ISO-BDM = 5, ISO-pAB = 5.

c. The *snai1b*⁺ CMs are not clearly visible in the control in Figure 2, panel (a).

d. In Figure 2, panel (c), it would make the Figure more impactful if we could see the *snai1b*-EGFP⁺ CM contribution to the trabecular layer in untreated 14 dpf hearts.

We appreciate the reviewer's comments and suggestions. Our revised **Figure 1d** (see our response to the comment above) enhances the visualization of the *snai1b*⁺ CMs distribution at 4 dpf (96 hpf). We have revised **Figure 2a-b** (see below) to include the comparison of *snai1b*-EGFP⁺ CMs in control vs. ISO-treated hearts at 14 dpf.

Figure 2.

(a-b) At 14 dpf, the expression of myocardial *snai1b* (dashed green outline) remained sparse in the ventricle, whereas Isoproterenol (ISO) treatment from 1 to 11 dpf revealed a persistent *snai1b* activation in the trabecular network.

Anatomic labels: BA, bulbus arteriosus; V, ventricle; BV, bulbus-ventricular annulus; Myo, myocardium; Endo, endocardium; Epi, epicardium; Ery, erythrocyte.

e. In Figure 3, panel (a), the representative images for the ventricle show a difference in chamber size. In panel (c), the representative image for control heart appears to have more complex trabeculation than the ISO treated. Are these observations consistent in all ISO treated hearts? A quantification of ventricular chamber size, total CM number, and trabecular CM number will help clarify this.

We thank the reviewer for their insightful comments and suggestions. In the revised **Figure S3** (see below), we showed that ISO-treatment did not change the number of compact and trabecular CMs, or the size of compact and trabecular cross-sectional area,

indicating that the overall development of the ventricle was unaffected. However, the size of the ventricle increased, and thinning of the trabecular, not compact, layer was observed, suggesting an increase in trabecular wall stress (Klabunde, 2021, 3rd edition). The control heart appears to have more complex trabeculation because it is less stretched. The activation of *snai1b* thus may be a compensating mechanism to generate more trabecular CMs to reduce the wall stress, as previously reported (Cairelli *et al.*, 2024 - PMID: 38345870).

Supplementary Figure 3. ISO-induced alteration in ventricular compact vs. trabecular morphology. Related to Figure 2.

(a) Representative images of control vs. ISO-treated *Tg(myl7:mKate-CAAX)* hearts at 4 dpf with nuclei labeled by DAPI. The right panels outline the morphology of compact and trabecular layers.

(b-e) ISO significantly increased the ventricle's cross-sectional area but not the number of either compact or trabecular CMs, as quantified using both *Tg(myl7:mCherry)* and *Tg(myl7:mKate-CAAX)* hearts.

(f-g) ISO did not change the absolute size of either the compact or trabecular layer but significantly reduced the thickness of the trabecular layer, while the thickness of the compact layer remained the same.

Six control and Six ISO-treated hearts were analyzed for each of the two reporter lines. All values are displayed with mean and standard error of mean (SEM). *p*-value is displayed for each comparison.

f. The hypothesis that *snai1b* is expressed in delaminating CMs as a result of strain variation/tension heterogeneity needs to also be verified during the physiological delamination process due to tension heterogeneity between 54-65 hpf window.

We thank the reviewer for their insightful suggestions. In the revised **Supplementary Figure 5b-e** (see below), we quantified the changes in the cross-sectional area using a *Tg(myl7:mKate-CAAX)* membrane line as CMs contracted at both 56 hpf and 96 hpf. Our data showed that ISO increased the deformation of trabecular but not compact CMs at both time points.

Supplementary Figure 5.

(b-e) 2-D cross-sections of the trabeculae (asterisks) reveal that the cross-sectional area of both compact (red dashed outlines) and trabecular (yellow dashed outlines) CMs enlarges during systole. At 56 hpf (c) and 96 hpf (e), ISO treatment significantly increased the enlargement of the area within trabecular CMs, whereas the enlargement in the compact CMs remained similar. All values are displayed with mean and standard error of mean (SEM). p -value is displayed for each comparison.

Anatomic labels: V, ventricle; A, atrium; AV, AV canal. One control heart and one ISO-treated heart were used for analysis at 56 hpf. Two control hearts and two ISO-treated hearts were used for analysis at 96 hpf.

g. The authors claim an average of 1.3 *snai1b*⁺ CM in a control 4 dpf heart, but Figure 4, panel (a) is not representative of the plot in Figure 4, panel (e).

We thank the reviewer for their insightful comments. The main purpose of this figure is to demonstrate that most *snai1b*⁺ CMs were Notch⁻, which requires high-magnification images. *snai1b*⁺ CMs are dispersed throughout the ventricle, and it is challenging to find a cluster of many *snai1b*⁺ CMs in the field of view under high magnification. If we highlight *snai1b*⁺ CMs in Figure 4a, it would be hard for readers to grasp the difference between panels (a) and (b)/(c). Therefore, we believe showing none is a better representation of the sparse *snai1b* expression in control 4 dpf hearts. We also revised the Y-axis title to “**Number of total ventricular *snai1b*⁺ CMs**” to avoid confusion.

h. In Figure 4, using a membrane line for marking the cardiomyocytes will help with getting resolution on marking and counting compact, delaminating, and trabecular CMs.

We thank the reviewer for their insightful suggestions. The current counting is based on *snai1b*⁺ RNAscope signal colocalizing with DAPI⁺ nuclei and *myl7*⁺ cytoplasm, which is more accurate than counting using the signal from the membrane, such as *Tg(myf7:mKate-CAAX)* in **Supplementary Figure 3** (see our response to an earlier comment). We found that membranes of neighboring CMs often intertwine like jigsaw puzzles at the border, rendering it challenging to parse out the exact number of cells.

i. In Figure 5, the authors intend to look at number of CMs undergoing delamination to form trabeculae upon activation/repression of *snai1b*. In physiological conditions, the most active window for observing delamination lies prior to 4dpf. An experiment checking for earlier time points will help with acquiring precise number of changes in delaminating CMs.

We thank the reviewer for their insightful suggestions. Our mosaic CRISPRa/i system usually targets only a small percentage of CMs and allows the overall development of the heart to be unperturbed. Therefore, it is more accurate to quantify “delaminated” CMs at 4 dpf than an earlier timepoint, as it is not easy to determine whether one CM is undergoing delamination. In the revised **Supplementary Figure 9b**, we corrected the terminology and, in **Figure 5c**, we provided a new quantification of trabecular CMs (see below). We also included additional experimental groups: Ctrl – no ISO, Repression – no ISO, and Activation – ISO.

Supplementary Figure 9.

(b) Percentage of delaminated (trabecular and apical) CMs in each ventricle across conditions. ISO did not significantly alter the delamination of CMs within the same CRISPR*a/i* groups. However, *snai1b*-repression under ISO treatment led to 37.5% of CMs undergoing apical delamination and 6.7% trabeculation, compared to 41.7% undergoing trabeculation in repression-only hearts (Figure 5d). All values are displayed with mean and standard error of mean (SEM). *p*-value is displayed for each comparison. Number of hearts analyzed: Control = 7, Repression = 11, Activation = 18.

Figure 5.

(b-d) Under ISO treatment, the majority of control (58.3%) and *snai1b*-repressed (56.7%) CMs remained in the compact layer at 4 dpf (96 hpf). Activation of *snai1b* led to significantly more delaminated (see Supplementary Figure 9b) and trabecular CMs per heart **(c)**, where 51.6% of total *snai1b*-activated CMs form trabeculae, compared to 25.0% of control and 6.7% of *snai1b*-repressed CMs **(d)**. On the other hand, repression of *snai1b* resulted in 36.7% of CMs undergoing apical delamination. For representative images of CRISPR*a/i*-injected hearts without ISO, see Supplementary Figure 9a. All values in panel c are displayed with mean and standard error of mean (SEM). *p*-value is displayed for each comparison. Number of hearts analyzed: Control-ISO = 7, Repression-ISO = 7, Activation-ISO = 11.

j. The authors claim that *snai1b*⁺ CMs are not regulated by canonical EMT pathways, but do not provide any data about the actual regulatory mechanism.

We thank the reviewer for their insightful comments. The upstream signaling genes between mechanical forces and *snai1b* are difficult to pinpoint as many signaling pathways converge to *snai1b*, which in turn regulates a multitude of downstream genes (Barrallo-Gimeno and Nieto, 2005 - PMID: 15983400). In the scope of the current paper, we have not yet deciphered the exact upstream mechanism.

Besides TGF- β , ErbB2, and Notch, we also looked at *piezo1* mRNA expression, but we did not observe differential expression related to *snai1b* in CMs (data not shown). In the current manuscript, we focused on revealing the mechanical activation of *snai1b* in CMs and the effect on myocardial delamination for trabeculation, both novel aspects of Snail genes. We hope our work laid the foundation for future exploration of the upstream non-canonical pathways.

Reviewer #2 (Remarks to the Author):

This research explores the role of the *snai1b* gene in the early stages of heart development, specifically in the process of trabeculation, where the heart muscle forms ridges essential for heart function.

Mechanics Comments

1. For the strain quantification in Figure 2d-g:

a. Please clarify what type of images were used for strain analysis analyzed, so as to verify that strains are indeed analyzed at the myocardial, instead of, say, the endocardium.

We thank the reviewer's insightful suggestions. We have included the following description in the main text: "At 4 dpf, we performed 4-D strain mapping on the *in vivo* images of *Tg(myl7:mCherry)* hearts acquired via spinning disk confocal microscopy." In the method section, we also detailed the specifications. The use of *myl7* reporter line ensures the fluorescent signal labels only the myocardium.

b. Will quantifying strains for the inner surface alone be misleading? ideally, the mid-wall surface, cutting right through the myocardium, rather than on either the epicardial or luminal side of the ventricular wall. This is so that we can be sure that strain measurements are done within the target tissue. Further, the inner surface of the heart will be very undulating with trabeculation structure, and strains can be hard to extract, especially when neighbouring trabecular structure touch during systole and become fused together in the reconstruction, causing strain errors. How is this managed? From this aspect, it seems that the inner surface may have more strain artefacts than the outer, smoother surface.

We thank the reviewer for their insightful comments and suggestions. In our revised **Figure 2c-e** (see our response to the next comment), we plotted the average strain and strain variation for **both the inner (endo) and outer (epi) surfaces** of the myocardium.

We agree that it is a common practice to calculate the mid-wall strain in ultrasound strain analysis. However, in our study, we aim to explore the differential strain in compact vs. trabecular layers, containing 1-2 sheets of cells at 4 dpf. Demonstrating the mid-wall strain would be more biologically relevant for a thick myocardium in adult hearts.

For our zebrafish larva heart, we characterize the epicardial strain to estimate the strain across the compact layer and the endocardial strain for the trabecular layer. The resulting

data indicated an increase in strain across ISO-treated endocardial surfaces but not ISO-treated epicardial surfaces (see **Figure 2d below**). The use of spinning disk confocal further enhanced the spatial and temporal resolution to minimize motion artifacts.

c. A worrying sign of high errors in the strain quantification is that in Fig 2g, the strain variations (explained as max areal strain – min areal strain) does not go down towards zero at the end of the cardiac cycle, when the heart should have returned to its original shape. As Fig 2g is used as a primary evidence for increased spatial variability of strains, this issue may impact the credibility of the claim. It would be good to see the authors’ validation of the strain quantification codes.

We thank the reviewer for their insightful comments and suggestions. In the revised **Figure 2c-e** (see below), we used a validated registration method that fits B-splines of Fourier to regularize the reconstructed cardiac motion (Wiputra *et al.*, 2020 - PMID: 33116206). We also used standard deviation to quantify strain variation, instead of the previous (max strain - min strain). The resulting strain variation returns to zero since the registration algorithm enforces the return of the heart to its starting shape.

Our previous registration program runs similarly, but it does not enforce periodic motion. Instead, it only use the relationship between the two adjacent time points to compute the deformation. As a result, small misalignments from the 4D synchronization step led to spots that failed to return to their original state. Nevertheless, it did not affect the periodicity of average strain, or the conclusion that ISO increased spatial variation in strain across both epi and endo surfaces.

Figure 2.

(c) At 4 dpf, 4-D mapping of myocardial strain in a control and an ISO-treated heart. Two time points during diastole were displayed, and the ISO-treated heart experienced higher strain at the end-diastole time point than the control. Red dashed lines and squares mark the cross-section planes, and the red arrows indicate the viewing direction. Blue and green arrows indicate the direction of blood flow.

Anatomic labels: V, ventricle; AV, atrioventricular canal; OFT, outflow tract.

(d) Average epicardial/endocardial ventricular strain within the sampling regions during one cardiac cycle. ISO treatment increased the myocardial strain across the endocardial surface during the end-diastolic and systolic phases.

(e) Strain variations within the sampling regions, calculated as the standard deviation of strains at each time point (phase). ISO treatment induced greater myocardial strain variations compared with the control heart.

d. Perhaps instead of claiming spatial variability – which I think may be very difficult, the authors should tone down to claim that ISO hearts generally contract and deform more, and leave the link between a greater deformation and greater spatial variability as a unproved and logical speculation.

We thank the reviewer for their insightful suggestions. We agree that the concept of strain variation during trabeculation is novel and has only recently been established by Priya *et al.*, 2020 (PMID: 33208950) and Cairelli *et al.*, 2024 (PMID: 38345870). These studies demonstrated that trabecular CMs experience higher strain along its fiber direction than the compact CMs. In our study, we aim to understand how ISO preferentially increased the strain within trabecular CMs that may have patterned the preferential activation of *snai1b* within trabecular (or pre-trabecular, Notch⁻ compact) CMs.

Our claim aligns with the previously reported findings and is supported by the molecular mechanism underlying ISO-activated actin-myosin interactions. As trabecular CMs develop more a mature actomyosin network than compact CMs, they would also be more responsive to ISO. From Figure 2 to Figure 3 and their supplementary figures, we demonstrate a path with increasing spatial resolution (4D walls > 3D ridges > 2D cells) to reach to our conclusion.

2. The division of the trabeculation into radial and circumferential may be contentious, some of the yellow lines in figure 3 do not seem to point to the AVC but are longitudinal in nature (apex – outflow tract in orientation, vertical in the images), but they are considered to be radial, one of the pink lines seem to be very close to the “left-right” axis, but is not considered radial. What is the specific definition of radial and what is the deviation tolerance of the definition? A concern is that this may be an unnatural pick-and-choose approach to generate statistical significance.

We thank the reviewer for their insightful comments and suggestions. We identify ridges that point towards the AV canal region as radial. In the revised **Figure 3a-b**, we corrected the radial ridges that did not point to the AV canal region as circumferential (see below, marked by asterisks*).

There is currently no convenient tool to isolate and categorize these trabecular ridges automatically. In future work, we can attempt to mathematically conceptualize the task and train a machine-learning model for automatic segmentation. We also attempted to manually isolate the ridges using the *Tg(myl7:mKate-CAAX)* membrane line (see our response to the following comment), but the membrane outline was much thinner than the cytoplasm and did not lead to satisfactory 4D reconstruction.

Figure 3a-b.

3. The trabeculation will naturally deform more than compact layer, as rightly pointed out by authors, this is due to myofiber alignment, but it may also be because the trabeculation has room to thicken, while compact layer is crowded and does not. My questions are:

a. The authors claim that because trabecular thickening is statistically significant with and without ISO while compact layer shortening is not, this means that there is higher spatial strain variation with ISO – this is not easy to accept. It is easy to imagine many scenarios where the observation holds true but spatial variation is not increased. For example, strains are proportionally increased with ISO in the same way everywhere, but measurement errors cause the compact layer data to be statistically insignificant due to its lower magnitudes. The authors should directly quantify spatial variation in strain if they wish to make this claim.

We thank the reviewer for their insightful comments. In **Figure 3c-d**, we chose to measure the shortening and expansion of CMs because the *Tg(myl7:mCherry)* reporter line labels the cytoplasm of CMs, not allowing us to see the precise borders of individual cells. We agree that, ideally, one should systematically analyze the 3D tissue strain within each trabecular and compact CMs. For this purpose, we attempted to perform 4D imaging on the *Tg(myl7:mKate-CAAX)* membrane line. However, we were not able to obtain accurate 3D outlines of individual cells after synchronization due to resolution issues. In addition, it is challenging to properly analyze the strain within an irregular 3D-shaped CM. Methods like Laplace-Dirichlet simulations are often used to determine local axes onto which the 3D Green-Lagrange strain can be projected. However, the axes usually fail to align with the anatomical directions of trabecular ridges, leading to biases.

During revision, we decided to measure the change in the cross-sectional area during contraction to further support our measurement of trabecular thickening (**Supplementary Figure 5b-e**, see below). The change in the cross-sectional area can be an estimate of axial strain because CMs are considered incompressible with a conserved volume during contraction (Fung, 1993, 2nd edition). Our data showed that ISO increased the change in the cross-sectional area during contraction in trabecular, but not compact, CMs at both 56 and 96 hpf. This suggests an increased difference in strain between compact and trabecular CMs (i.e., increased spatial variation).

We also agree that the difference in deformation may arise from compact CMs not having room to thicken. On the other hand, that does not conflict with our claim that the compact CMs would experience a smaller increase in strain after ISO treatment, due to the conservation of volume. Our future investigation will need to optimize our imaging and modeling methods to further study this phenomenon. In the current study, we focus on the mechanical activation of *snai1b* in CMs and the effect on myocardial delamination for trabeculation.

Supplementary Figure 5.

(b-e) 2-D cross-sections of the trabeculae (asterisks) reveal that the cross-sectional area of both compact (red dashed outlines) and trabecular (yellow dashed outlines) CMs enlarges during systole. At 56 hpf **(c)** and 96 hpf **(e)**, ISO treatment significantly increased the enlargement of the area within trabecular CMs, whereas the enlargement in the compact CMs remained similar. *p*-value is displayed for each comparison.

Anatomic labels: V, ventricle; A, atrium; AV, AV canal. One control heart and one ISO-treated heart were used for analysis at 56 hpf. Two control hearts and two ISO-treated hearts were used for analysis at 96 hpf.

b. It is also not clear why the difference between the trabecular and compact layer strain is the main point here. To be relevant to trabeculation formation and delamination, we really need to look for localized variation in strain between neighbouring cells from the same layer in the pre-trabeculated myocardial wall. After the trabeculation is formed, there will naturally be differences in strains between trabeculation and compact layers, and it seems that this trabecular-compact strain difference will not contribute to any further trabecular tissue formation. It seems that the wrong type of spatial strain variation is used to substantiate a claim that there is a pro-trabeculation biomechanical environment.

We thank the reviewer for their insightful suggestions. In this study, our focus is to understand how ISO preferentially increased the strain within trabecular CMs that may pattern the preferential activation of *snai1b* within trabecular (or pre-trabecular Notch-compact) CMs. We do not seek to study a broad phenomenon of spatial strain variation leading to trabeculation, as previously reported (Priya *et al.*, 2020 - PMID: 33208950; Cairelli *et al.*, 2024 - PMID: 38345870).

Moreover, clonal analysis of zebrafish CMs has shown that the trabecular fibers at 10 dpf contain CMs of heterogenous origin (Gupta and Poss, 2012 - PMID: 22538609). CM delamination has also been observed between 80 and 90 hpf (Jimenez-Amilburu *et al.*, 2016 - PMID: 27926871). These studies suggest that the delamination of CMs likely occurs beyond 56 hpf, and trabecular-compact strain difference should continue to modulate the process.

Other Thoughts:

1. If *snai1b*+ CM is so sparse, only 1-2 cells per heart, this does not seem enough to support the large areas of trabeculation occurring in the heart. Does this mean that *snai1b* is not important to natural trabeculation?

We thank the reviewer's question. *snai1b* is important for early-stage cardiogenesis prior to trabeculation, as previously reported (Qiao *et al.*, 2015 - PMID: 24667151 and Gentile *et al.*, 2021 - PMID: 34152269). While we observed that *snai1b* is not widely expressed during natural trabeculation, we uncovered that *snai1b* is responsive to elevated myocardial strain and is implicated in coordinating the initiation of delamination for trabecular CMs.

As demonstrated in our revised **Figure 5c-d** (see below), without ISO, *snai1b* repression did not affect the percentage of trabecular CMs (41.7%), whereas with ISO treatment, it significantly reduced the percentage of trabecular CMs (6.7%). Moreover, 36.7% of *snai1b*-repressed CMs delaminated apically. Our data showcased the importance of *snai1b* in guiding the proper delamination of CMs. Finally, *snai1b* may also be tightly regulated and activated in a stochastic fashion among CMs.

Figure 5

(c-d) Under ISO treatment, the majority of control (58.3%) and *snai1b*-repressed (56.7%) CMs remained in the compact layer at 4 dpf (96 hpf). Activation of *snai1b* led to significantly more delaminated (see Supplementary Figure 9b) and trabecular CMs per heart **(c)**, where 51.6% of total *snai1b*-activated CMs form trabeculae, compared to 25.0% of control and 6.7% of *snai1b*-repressed CMs **(d)**. On the other hand, repression of *snai1b* resulted in 36.7% of CMs undergoing apical delamination. For representative images of CRISPRa/i-injected hearts without ISO, see Supplementary Figure 9a. All values in panel c are displayed with mean and standard error of mean (SEM). *p*-value is displayed for each comparison. Number of hearts analyzed: Control-ISO = 7, Repression-ISO = 7, Activation-ISO = 11.

2. It seems good to have a basic demonstration of which condition will result in more trabeculation and which will have less, via plain microscopy imaging, and quantification of trabeculation tissue area/volume. This is not so easy to appreciate from images with very busy fluorescent labels (eg., fig 5).

We thank the reviewer for their insightful suggestions. In the revised **Supplementary Figure 3** (see our response to an earlier comment), we quantified the number of compact and trabecular CMs, as well as the size of compact and trabecular cross-sectional areas.

For the CRISPRa/i experiment in Figure 5, we do not expect overall trabeculation to be different across the groups, as the mosaic system only targets a small percentage of CMs. It allows for investigating the effect of *snai1b* activation and repression without changing the overall cardiac function.

3. The authors did not show a comparison of *snai1b*+ with wildtype without ISO, it seems that such a baseline control will be important.

We thank the reviewer for their insightful suggestions. Although it is unclear which figure the reviewer is referring to, we provided control data in revised **Figure 2a** (for 14 dpf *snai1b* expression) and **Supplementary Figure 9a** (for control-no ISO, quantification see **Figure 5c-d** and **Supplementary Figure 9b**).

Figure 2.

(a-b) At 14 dpf, the expression of myocardial *snai1b* (dashed green outline) remained sparse in the ventricle, whereas Isoproterenol (ISO) treatment from 1 to 11 dpf revealed a persistent *snai1b* activation in the trabecular network.

Anatomic labels: BA, bulbus arteriosus; V, ventricle; BV, bulbus-ventricular annulus; Myo, myocardium; Endo, endocardium; Epi, epicardium; Ery, erythrocyte.

Supplementary Figure 9.

(a) Representative images of CRISPRa/i-injected hearts at 4 dpf (96 hpf) without ISO.

(b) Percentage of delaminated (trabecular and apical) CMs in each ventricle across conditions. ISO did not significantly alter the delamination of CMs within the same CRISPRa/i groups. However, *snai1b*-repression under ISO treatment led to 37.5% of CMs undergoing apical delamination and 6.7% trabeculation, compared to 41.7% undergoing trabeculation in repression-only hearts (Figure 5d). All values are displayed with mean and standard error of mean (SEM). *p*-value is displayed for each comparison. Number of hearts analyzed: Control = 7, Repression = 11, Activation = 18.

Anatomic labels: V, ventricle; Comp, compact layer; Trab, trabeculae.

Figure 5

(c-d) Under ISO treatment, the majority of control (58.3%) and *snai1b*-repressed (56.7%) CMs remained in the compact layer at 4 dpf (96 hpf). Activation of *snai1b* led to significantly more delaminated (see Supplementary Figure 9b) and trabecular CMs per heart (c), where 51.6% of total *snai1b*-activated CMs form trabeculae, compared to 25.0% of control and 6.7% of *snai1b*-repressed CMs (d). On the other hand, repression of *snai1b* resulted in 36.7% of CMs undergoing apical delamination. For representative images of CRISPRa/i-injected hearts without ISO, see Supplementary Figure 9a. All values in panel c are displayed with mean and standard error of mean (SEM). *p*-value is displayed for each comparison. Number of hearts analyzed: Control-ISO = 7, Repression-ISO = 7, Activation-ISO = 11.

Biological comments:

1. The quantification of snail activation (Fig.1a-b) is confusing. The authors should include the number of cells positive along with the number of animals analysed. They should also quantify where the ectopic cells activating Snail are located.

We thank the reviewer's insightful suggestion. We have included the quantification of *snai1b*-EGFP⁺ and *snai1b*-mRNA⁺ CMs from 56 hpf to 96 hpf in the revised **Figure 1e** (see below). The number of hearts analyzed for each group is indicated in **Figure 1d**. We quantified the percentage of *snai1b*⁺ CMs present in compact vs. trabecular layers in **Figure 4f** (see below), where 62.5% of control and 50% of ISO-treated *snai1b*⁺ CMs are trabecular.

Figure 1.

(d) During trabeculation (56 hpf – 96 hpf), *snai1b:EGFP* signal is concentrated around the BV annulus. Isoproterenol (ISO) treatment at 1 dpf induced *snai1b* activation in the ventricular CMs.

(e) Whole-mount *in situ* hybridization of *snai1b* mRNA in the *Tg(snai1b:EGFP; myl7:mCherry)* reporter line reveals a significant increase of both EGFP⁺ and mRNA⁺ CMs in ISO-treated hearts. (see Supplementary Figure 1-2 for staining images). All values are displayed with mean and standard error of mean (SEM). *p*-value is displayed for each comparison. Number of hearts analyzed in each is displayed in panel d.

Anatomic labels: BA, bulbus arteriosus; V, ventricle; BV, bulbus-ventricular annulus; AV, atrioventricular canal.

Figure 4f.

2. Isoproterenol treatments: the treatment starts at 1 dpf and heart are analysed at 4dpf or 14dpf. The authors never show if the treatment is efficient in activating strain throughout the length of the treatment. I would think that the effect of the drug will vanish with time. A possibility is that the early treatment leads to abnormal development of the heart leading to aberrant force distribution in the contracting heart, thus independent from the change in strain. The authors need to clarify this.

We thank the reviewer's insightful comments and suggestions. In the revised **Supplementary Figure 5a** (see below), we provided the heart rate and ejection fraction data from 48 hpf to 96 hpf of the same group of embryos, demonstrating the persistent effect of ISO during trabeculation. Unfortunately, larvae older than 5 dpf were no longer transparent for optical imaging and too small for ultrasound imaging to provide functional data.

We agree that chronic, high-dose ISO treatment leads to desensitization (Kossact *et al.*, 2017 - PMID: 2854511). However, our ISO dosage (100 mM) is 10 times lower than the reported study. For ISO treatment past 5 dpf, the larvae were raised in the core facility during the day (12 hours) and kept in ISO solution overnight (12 hours) to ensure proper feeding. We have updated the method section with this detail.

In the revised **Supplementary Figure 3** (see below), we revealed that ISO-treatment did not change the number of compact and trabecular CMs, or the size of compact and trabecular cross-sectional area, indicating that the overall development of ventricle was unaffected. However, the size of ventricle increased (**Supplementary Figure 3b, d**), and the thinning of trabecular, not compact, layer was observed (**Supplementary Figure 3g**), suggesting an increase in trabecular wall stress (Klabunde, 2021, 3rd edition).

Finally, in the revised **Supplementary Figure 4c** (see below), we revealed that treating the hearts at a later stage (3 dpf) for 24 hours is sufficient to activate the transcription of *snai1b* mRNA, and co-treatment with myosin inhibitors blunted the activation. At this stage, the overall development of the heart was unlikely to be affected within 24 hours, suggesting that *snai1b* activation is caused by changes in strain.

Supplementary Figure 5. (a) ISO-treatment induced a significant increase in ejection fraction (EF) and heart rate (HR) from 48 hpf to 96 hpf. p -value is displayed for each comparison.

Supplementary Figure 3. ISO-induced alteration in ventricular compact vs. trabecular morphology. Related to Figure 2.

(a) Representative images of control vs. ISO-treated *Tg(myl7:mKate-CAAX)* hearts at 4 dpf with nuclei labeled by DAPI. The right panels outline the morphology of compact and trabecular layers.

(b-e) ISO significantly increased the ventricle's cross-sectional area but not the number of either compact or trabecular CMs, as quantified using both *Tg(myl7:mCherry)* and *Tg(myl7:mKate-CAAX)* hearts.

(f-g) ISO did not change the absolute size of either the compact or trabecular layer but significantly reduced the thickness of the trabecular layer, while the thickness of the compact layer remained the same.

Six control and Six ISO-treated hearts were analyzed for each of the two reporter lines. All values are displayed with mean and standard error of mean (SEM). p -value is displayed for each comparison.

Supplementary Figure 4. (c) ISO treatment starting at 72 hpf instead of 24 hpf reveals a statistically significant decrease in EGFP⁺ CMs, but not mRNA⁺ CMs, using the whole-mount *in situ* hybridization of *snai1b* mRNA in *Tg(snai1b:EGFP; myl7:mCherry)* hearts at 96 hpf. Applying pAB along with ISO at 72 hpf led to a significant decrease of mRNA⁺ CMs, compared to ISO alone. All values are displayed with mean and standard error of mean (SEM). p -value is displayed for each comparison. Number of hearts analyzed: ISO-24hpf = 6, ISO-72hpf = 7, ISO-BDM = 5, ISO-pAB = 5.

3. The cell deformation in response to strain described in figure 3 is not conclusive. How can you conclude without having a three dimensional analysis of the single cell? The change in shape could simply be due to an artefact due to a different plane of view.

We thank the reviewer's insightful comments. We agree that a 3D analysis on CMs would be the ideal approach. We attempted to perform 4D imaging on the *Tg(myl7:mKate-CAAX)* membrane line for this purpose. However, we were not able to obtain accurate 3D outlines of individual cells after synchronization, as we mentioned in an earlier response.

In revised **Supplementary Figure 5b-e** (see above), we decided to measure the change in the cross-sectional area during cardiac systole (contraction) to further support our measurement of trabecular thickening. The change in the cross-sectional area can be an estimate of axial strain because CMs are considered incompressible with a conserved volume during contraction (Fung, 1993, 2nd edition). On the other hand, 3D strain analysis also entails biases, such as the determination of local axes via Laplace-Dirichlet simulation.

We also agree that the movement of z-plane can be a source of error. We tried to minimize it by analyzing slices near the center long-axis plane, where most of the cells were engaging in-plane movements. We also took care to measure cells that appear in every

frame of the time course. We added the following in our discussion: “*Our current study performed 2D deformation analysis on trabecular and compact CMs due to the difficulty of obtaining precise borders of the cells. In the future, it would be plausible to optimize the imaging and segmentation pipeline for 3D tissue strain analysis within the individual CMs.*”

4. Fig 6 and 7 are not completely relevant to the study, not sure they belong to main figures.

We thank the reviewer’s comments. Figures 6 & 7 are to further support the notion that myocardial *snai1b* activation is unique compared with *snai1b* activation in endo-/epi-cardial cells. First, the scRNA seq dataset reflected the sparsity of *snai1b*⁺ CMs within the heart, while the *snai1b*⁺ ECs and EpCs can be readily observed from both the dataset and mRNA staining. Second, our data showed that myocardial *snai1b* is not involved in the canonical EndoMT/EMT/fibrosis pathways. Unfortunately, we have not yet deciphered the exact upstream mechanism.

Overall, our study is to demonstrate a previously unknown function of *snai1b*. We believe such data, combined with previous figures, can inspire readers to think beyond the conventional roles of *snai1b* and larger Snail family genes, as they are largely unrecognized as regulatory genes for myocardial development.

Reviewer #3 (Remarks to the Author):

This study explores the role of *snai1b*-positive cardiomyocytes (CMs) in the initiation of trabeculation during heart development in zebrafish. Unlike typical Snail family genes involved in epithelial-to-mesenchymal transition (EMT), these *snai1b*⁺ CMs are mechano-sensitive and initiate delamination for trabecular organization. At 4 days post-fertilization, *snai1b*⁺ CMs were sparse, but their numbers increased significantly when myocardial strain was enhanced using Isoproterenol (ISO). Most of these CMs were Notch-negative. CRISPR-activation of *snai1b* promoted trabeculation, while its repression reduced trabecular formation, even under strain. The study also found that *snai1b*⁺ CMs do not exhibit typical EMT markers, like collagen 1a2 production. These findings suggest that *snai1b*⁺ Notch-negative CMs are mechanically activated to drive cardiac trabeculation independently of traditional EMT pathways.

In this article, the authors describe how strain-induced *snai1b* expression in notch1-repressed cardiomyocytes elicits their delamination, the initial step of trabeculation. They demonstrate the increase in *snai1b*⁺ CM numbers using a reporter of *snai1b* that does not seem to recapitulate *snai1b* expression, neither in the epicardium and the endocardium, as expressed by the authors in lines 92-94, nor in the cardiomyocytes, contrary to their claim in those lines, as seen in Figure S1 and Fig4a. There, expression of *snai1b* is widespread in almost all cardiomyocytes. Thus, validity of this reporter in the context of the heart is not, to my view, sufficiently proved.

However, the RNA-seq reanalysis performed was a fruitful approach and the subsequent experiments on *snai1b*-mediated regulation of *col1a2* and TGF- β induction of *snai1b* expression, showing independency of this signals in the regulation of the delamination of CMs is surprising but very interesting.

Several experiments would be needed, specifically:

-To further validate the reporter in this system: mRNA expression analysis (as in Supp. Fig. 1 or in Fig. 4a) at different stages to compare with reporter data.

We thank the reviewer's insightful comments and suggestions. The widespread EGFP signal in **Supplementary Figure 1** and **Fig 4a** is TP1:EGFP (notch activity), but not *snai1b*.

To validate our reporter system, we performed *in situ* hybridization of *snai1b* mRNA in the *Tg(snai1b:EGFP; myl7:mCherry)* hearts from 56 hpf to 96 hpf and quantified the percentage of both EGFP⁺ and mRNA⁺ CMs in the revised **Supplementary Figure 2** (see below). The results corroborated our *snai1b* reporter activity in agreement with *snai1b* mRNA staining in cardiomyocytes.

Supplementary Figure 2. Whole-mount *in situ* hybridization of *snai1b* mRNA in the larval hearts of *Tg(snai1b:EGFP; myl7:mCherry)* reporter line. Related to Figure 1.

(a-d) Whole-mount *in situ* hybridization of *snai1b* mRNA (arrowheads) reveals that the reporter activity is myocardial-specific at 56 hpf **(a)**, 72 hpf **(b)**, 96 hpf **(c)**, and 5 dpf **(d)**.

(e) Percentage of *snai1b:EGFP*-positive cardiomyocytes (CMs) that are also positive for mRNA staining at 56 hpf (Ctrl n = 6, ISO n = 5), 72 hpf (Ctrl n = 7, ISO n = 7), and 96 hpf (Ctrl n = 6, ISO n = 6). All values are displayed with mean and standard error of mean (SEM). *p*-value is displayed for each comparison.

Anatomic labels: BA, bulbus arteriosus; V, ventricle; AV, atrioventricular canal; BV, bulbus-ventricular annulus; Myo, myocardium.

-To supplement the data in figure 4, maybe the opposite experiment, using Notch1 ICD IHC to show absence of colocalization with *snai1b*⁺ CMs, as data shown in Fig. 4 a-c do not correspond to quantification in e.

We thank the reviewer for their insightful suggestions. Zebrafish antibodies are often poorly validated, and the TP1 reporter is widely accepted in the field for visualizing notch signaling. In addition, demonstrating *snai1b*⁺/Notch⁻ CMs requires high-magnification images. *snai1b*⁺ CMs are dispersed throughout the ventricle, and it is difficult to find a cluster of many *snai1b*⁺ CMs in the field of view under high magnification. Performing Notch1 ICD IHC would not improve the image. Here, we highlight sparse *snai1b* expression in control hearts by showing no *snai1b*⁺ CMs in Figure 4a, so that readers can grasp the difference between control and ISO groups. We also revised the Y-axis title to “Number of total ventricular *snai1b*⁺ CMs” to avoid confusion.

-Figure 4, it would be useful to quantify the Notch⁺ CMs, as they seem to decrease in number when ErbB2 is inhibited.

We thank the reviewer for their insightful suggestions. In the revised **Supplementary Figure 7** (see below), we quantified the number of total ventricular Notch⁺ CMs in each group. The data shows that the co-treatment of ErbB2 inhibitor (PD) and ISO significantly reduced the number of Notch⁺ CMs, while ISO alone did not.

Supplementary Figure 7. Myocardial Notch activity in control vs. ISO-treated hearts. Related to Figure 4.

(a) Representative images of myocardial Notch activity in control, ISO-treated, and ISO-ErbB2 inhibitor-cotreated hearts at 96 hpf. (b) ErbB2 inhibitor (PD) significantly reduced the total number of Notch⁺ CMs per ventricle, while ISO alone did not. Number of hearts analyzed: control n = 6; ISO n = 7; ISO+PD@72hpf n = 7; ISO+PD@55hpf n = 5. All values are displayed with mean and standard error of mean (SEM). *p*-value is displayed for each comparison.

-In line 207, related to data shown in Figure 4, the authors show a decrease in trabeculation when ErbB2 is inhibited. But there is no quantitative data and panels shown in Suppl. Fig. 3 are not that clear. Can they provide these quantitative data?

We thank the reviewer for their suggestions. In the revised **Supplementary Figure 6e** (see below), we quantified the area of compact vs. trabecular layers in each group.

Supplementary Figure 6.

(d-e) Treating with ErbB2 inhibitor, PD168393, from 55 hpf or 72hpf significantly reduced the area of trabeculae at 4 dpf. 5 hearts were analyzed for each condition. Number of hearts analyzed: control n = 6; ISO n = 7; ISO+PD@72hpf n = 7; ISO+PD@55hpf n = 5. All values are displayed with mean and standard error of mean (SEM). *p*-value is displayed for each comparison. Anatomic labels: V, ventricle; A, atrium.

-In figure 5 an appropriate explanation of why the *snai1b*-activated fish were not subject to ISO treatment is necessary, as well as the comparison with *snai1b* repressed without isoproterenol.

We thank the reviewer for their insightful suggestions. In the original figure, we aimed to demonstrate that *snai1b* activation led to more delaminated and trabecular CMs. Therefore, we removed ISO to prevent contribution from mechanical forces. In the revised **Figure 5b-d** and **Supplementary Figure 9a-b**, we included additional groups: Ctrl – no ISO, Repression – no ISO, and Activation – ISO (see below). The results show that *snai1b*-activation led to a significant increase in the percentage of delaminated and trabecular CMs with or without ISO.

Figure 5.

(b-d) Under ISO treatment, the majority of control (58.3%) and *snai1b*-repressed (56.7%) CMs remained in the compact layer at 4 dpf (96 hpf). Activation of *snai1b* led to significantly more delaminated (see Supplementary Figure 9b) and trabecular CMs per heart **(c)**, where 51.6% of total *snai1b*-activated CMs form trabeculae, compared to 25.0% of control and 6.7% of *snai1b*-repressed CMs **(d)**. On the other hand, repression of *snai1b* resulted in 36.7% of CMs undergoing apical delamination. For representative images of CRISPRa/i-injected hearts without ISO, see Supplementary Figure 9a. All values in panel c are displayed with mean and standard error of mean (SEM). *p*-value is displayed for each comparison. Number of hearts analyzed: Control-ISO = 7, Repression-ISO = 7, Activation-ISO = 11. Anatomic labels: V, ventricle; Comp, compact layer; Trab, trabeculae.

Supplementary Figure 9.

(a) Representative images of CRISPRa/i-injected hearts at 4 dpf (96 hpf) without ISO.

(b) Percentage of delaminated (trabecular and apical) CMs in each ventricle across conditions. ISO did not significantly alter the delamination of CMs within the same CRISPRa/i groups. However, *snai1b*-repression under ISO treatment led to 37.5% of CMs undergoing apical delamination and 6.7% trabeculation, compared to 41.7% undergoing trabeculation in repression-only hearts (Figure 5d). All values are displayed with mean and standard error of mean (SEM). *p*-value is displayed for each comparison. Number of hearts analyzed: Control = 7, Repression = 11, Activation = 18.

Anatomic labels: V, ventricle; Comp, compact layer; Trab, trabeculae.

-Fig. 5c refers to percentage of delaminating CMs/cerulean + in KRAB? Which normalization is then used for VPR? Otherwise, is it divided by the total number of CMs?

We thank the reviewer's questions. In the revised **Figure 5c** and **Supplementary Figure 9b**, we quantified the percentage of trabecular or delaminated CMs in each ventricle. The number was divided by the total number of CRISPRa/i-labeled CMs.

-The claim that *snai1b* repressed cells that "delaminated apically" are CMs is not properly substantiated, as they do not show any CM identity marker, besides the fact that *myl7* is driving the expression of cerulean. But in suppl. Fig. 4c there are Cerulean+ cells that are not mCherry+, despite having the same transgenic strategy. In order to ensure that those cells are CMs, it would be interesting to have any cardiomyocyte marker analysed either by ISH or IHC.

We thank the reviewer for their insightful suggestions. The “cerulean+” cells that are not *myl7*⁺ in the original Figure S4c (now S8c) were caused by autofluorescence from blood cells. The apical delamination of CMs has been documented in *snai1b* KO hearts at 50 hpf (Gentile *et al.*, 2021 - PMID: 34152269). As a result, we expected to see them in our experiments. In the revised **Supplementary Figure 9c**, we stained these CMs with a myocardial-specific marker, myosin 4 (MF20), to corroborate their identity.

Supplementary Figure 9.

(a) Staining of myocardial-specific marker, myosin 4 (MF20), on apically delaminated CMs (arrowhead).

-Regarding the scRNAseq data in figure 6, I think it would be informative to compare the *snai1b*⁺ CMs from the BV annulus to their *snai1b*⁻ counterparts in the same tissue, as their contribution when comparing with ventricular CMs (which was the case, I think) might be minor (they represent only 27% of the CMs).

We thank the reviewer for their insightful suggestions. We agree that comparing CMs within the same compartment would be the ideal approach. However, the GO analysis was not successful due to the small number of *snai1b*⁺ CMs. We therefore compared BV vs. ventricular CMs as a surrogate, yielding more meaningful GO results and allowing for continuing RNAScope for validation.

-Moreover, I think that, a scRNAseq experiment with ISO treated hearts can produce more useful information.

We respectfully agree with the reviewer. The contemporary approach to isolate sufficient single cells from zebrafish embryonic hearts remains a tremendous experimental challenge. In the dataset we reanalyzed, the authors dissected >1000 hearts from 5-dpf embryos to collect 366 post-filtering single cells (Weinberger *et al.*, 2020 - PMID: 32084358). It would be fitting for a future study to focus on the upstream and downstream pathways of myocardial *snai1b*. In the current manuscript, we focused on revealing the mechanical activation of *snai1b* in CMs and the effect on myocardial delamination for trabeculation, both novel aspects of Snail genes. We hope our work laid the foundation to conduct the sc RNA-seq experiments in our future investigation.

RESPONSE TO REVIEWERS' COMMENTS

Reviewer #2 (Remarks to the Author):

The authors have addressed all the points, well done for this very interesting study.

We would like to express our sincere appreciation to the reviewer for their recognition of our study and the many constructive suggestions.

Reviewer #3 (Remarks to the Author):

This revised manuscript presents an insightful and well-executed study identifying a novel role for mechanically activated *snai1b*⁺ cardiomyocytes in initiating myocardial delamination and trabeculation in the developing zebrafish heart. The authors convincingly show that *snai1b* expression is induced by myocardial strain, independently of hemodynamic shear stress, and functions upstream of trabecular morphogenesis through a distinct, Notch-negative cell population. These findings represent a significant contribution to the understanding of cardiac mechanotransduction and ventricular wall patterning.

Strengths of the study include the combination of CRISPRa/i technology, 4D strain mapping, and single-cell/spatial transcriptomics, which together provide a robust and multidimensional view of *snai1b*'s role. The data are well-presented, and the experimental approach is methodologically sound. Importantly, the identification of *snai1b*⁺ CMs that diverge from canonical EMT and are selectively sensitive to mechanical cues advances the field beyond prior models reliant on shear stress or Notch signaling alone.

The revised version of this manuscript has improved it substantially and has answered my previous concerns and suggestions.

We would like to express our sincere appreciation to the reviewer for acknowledging the validity and novelty of our study. We also thank them for the suggestions about improving our manuscripts.